# Personalized Subgraph Federated Learning

## Abstract

In real-world scenarios, subgraphs of a larger global graph may be distributed across multiple devices or institutions, and only locally accessible due to privacy restrictions, although there may be links between them. Recently proposed subgraph Federated Learning (FL) methods deal with those missing links across private local subgraphs while distributively training Graph Neural Networks (GNNs) on them. However, they have overlooked the inevitable heterogeneity among subgraphs, caused by subgraphs comprising different parts of a global graph. For example, a subgraph may belong to one of the communities within the larger global graph. A naive subgraph FL in such a case will collapse incompatible knowledge from local GNN models trained on heterogeneous graph distributions. To overcome such a limitation, we introduce a new subgraph FL problem, personalized subgraph FL, which focuses on the joint improvement of the interrelated local GNN models rather than learning a single global GNN model, and propose a novel framework, *FEDerated Personalized sUBgraph learning* (FED-PUB), to tackle it. A crucial challenge in personalized subgraph FL is that the server does not know which subgraph each client has. FED-PUB thus utilizes functional embeddings of the local GNNs using random graphs as inputs to compute similarities between them, and use them to perform weighted averaging for server-side aggregation. Further, it learns a personalized sparse mask at each client to select and update only the subgraph-relevant subset of the aggregated parameters. We validate FED-PUB for its subgraph FL performance on six datasets, considering both non-overlapping and overlapping subgraphs, on which ours largely outperforms relevant baselines.

## 1 Introduction

A graph, which defines the relationships among instances, can model a wide range of structured data including social [7], co-purchasing [23], and collaboration networks [36]. Most of the previous works on graph representation learning focus on a single graph, whose nodes and edges collected from multiple sources are stored in a central server. For instance, in a social network platform, every user, with his/her social networks, contributes to creating a giant network consisting of all users and their connections. However, in some practical scenarios, each user/institution collects its own private graph, which is only locally accessible due to privacy restrictions. For instance, as described in Zhang et al. [45], each hospital may have its own patient interaction network to track their physical contacts or co-diagnosis of a disease, however, such a graph may not be shared with others. An obvious challenge for such a scenario is how to deal with potentially missing edges between subgraphs [42, 45] that are not captured by individual data owners, that may carry important information (See Figure 1 (A)).

How can we then collaboratively train, without sharing actual data, a neural network with its subgraphs distributed across multiple participants (i.e., clients) over different devices or institutions? The most straightforward way is to perform Federated Learning (FL) with Graph Neural Networks (GNNs). In particular, in such an FL framework, each client will individually train a local GNN on the private

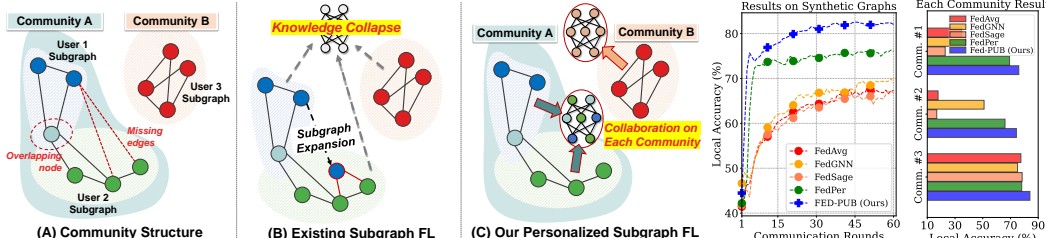

Figure 1: **(A) An illustration of local subgraphs distributed across multiple participants** with overlapping nodes, missing edges and community structures between subgraphs. **(B) Existing subgraph FL methods** [42, 45] expand the local subgraphs to tackle the missing edge problem, but collapse incompatible knowledge from heterogeneous subgraphs. **(C) Our personalized subgraph FL** focuses on the joint improvement of local models working on interrelated subgraphs, such as ones within the same community, by selectively sharing knowledge across them. **(Right:) Knowledge collapse results**, where local models belonging to two small communities (Comm 1 and 2) suffer from large performance degeneration by existing subgraph FL (e.g., FedGNN [42] and FedSage+ [45]). A personalized FL method, FedPer [2] also underperforms ours since it only focuses on individual model's improvement without sharing local personalization layers between similar subgraphs.

local data, while a central server aggregates the locally updated GNN weights from multiple clients into one, and then transmits it back to the clients. Recent subgraph FL methods work in such a manner [42, 45] while additionally tackling the problem of missing edges between subgraphs. This is done as illustrated in Figure 1 (B), where the local subgraph is expanded either by exactly augmenting the relevant nodes from the other subgraphs at the other clients [42], or by estimating the nodes using the node information in the other subgraphs [45]. However, such sharing of node information may compromise data privacy and can incur high communication costs.

Also, there exists a more important challenge that has been overlooked by the existing subgraph FL methods. We observe that they suffer from large performance degeneration (See Figure 1 right), due to a lack of consideration of the *heterogeneity* among the subgraphs, which is natural since subgraphs comprise different parts of a global graph. Notably, there could be multiple communities within a global graph, each of which is formed by a group of densely connected subgraphs with similar characteristics (Figure 1 (A)). For example, some of patient networks from hospitals can be grouped by their specialized sectors according to the disease categories, namely psychiatric or ophthalmology.

Motivated by this challenge, we introduce a novel problem of personalized subgraph FL, whose goal is the joint improvement of interrelated local models trained on the interconnected local subgraphs, for instance, subgraphs belonging to the same community (See Figure 1 (C)), by sharing weights among them. However, tackling personalized subgraph FL is challenging, since we do not know which subgraph each client has, due to their local accessibility. To resolve this issue, we use functional embeddings of GNNs on random graphs to obtain similarity scores between two local GNNs, inspired by a work for neural network search that effectively represents entire neural networks in the vector space [17], and then use them to perform weighted averaging of the model weights at the server. However, the similarity scores only tell how relevant each local model from the other clients is, but not which of the parameters are relevant. Thus we further learn and apply personalized sparse masks on the local GNN at each client to obtain only the subnetwork, relevant for the local subgraph. We refer to this subgraph FL framework as *FEDerated Personalized sUBgraph learning* (FED-PUB).

We extensively validate our FED-PUB on six different datasets with varying numbers of clients, under both overlapping and disjoint subgraph FL scenarios. The experimental results show that ours significantly outperforms relevant baselines. Further analysis shows that our method can discover community structures among subgraphs, and the subgraph-specific masking localizes the knowledge with respect to subgraphs belonging to each community. Our main contributions are as follows:

- We introduce a novel problem of personalized subgraph FL, which aims at collaborative improvements of the related local models (e.g. subgraphs belonging to the same community), which has been relatively overlooked by previous works on graph and subgraph FL.

- We propose a novel framework for personalized subgraph FL, which performs weighted averaging of the local model parameters based on their functional similarities obtained without accessing the data, and learns sparse masks to select only the relevant subnetworks for the given subgraphs.

- We validate our personalized subgraph FL framework on six real-world datasets under two different settings, demonstrating its effectiveness over existing subgraph FL baselines.

## 2 Related Work

**Graph Neural Networks**  Graph representation learning with Graph Neural Networks (GNNs) [10, 48, 43, 18, 3], which aims to learn the representations of the nodes, the edges, and the entire graph, is an extensively studied topic. Most existing GNNs under the message passing scheme [8] iteratively represent a node by aggregating features from its neighboring nodes as well as itself. For example, Graph Convolutional Network (GCN) [22] approximates the spectral graph convolutions [12], yielding a mean aggregation over neighboring nodes. Similarly, for each node, GraphSAGE [11] aggregates the features from its neighbors to update the node representation. Such advances in GNNs have led to successes on node and link prediction tasks [22, 47]. However, they are not directly applicable to real-world systems with locally distributed graphs, where graphs from different sources are not shared across participants, which gives rise to federated learning approaches to train GNNs.

**Federated Learning**  Federated Learning (FL) [32, 41, 19, 24], aiming to learn a model by aggregating model weights trained on local data, is an essential approach for our distributed subgraph learning problem. To mention a few, FedAvg [32] locally trains a model for each client and then transmits the trained model to a server, while the server aggregates the model weights from local clients and then sends the aggregated model back to them. However, since the locally collected data from different clients may largely vary, heterogeneity is a crucial issue. To tackle this, FedProx [25] proposes the regularization term that minimizes the weight differences between local and global models, which prevents the model from diverging by overfitting to the local training data. However, when the local data is extremely heterogeneous, it is more appropriate to collaboratively train a personalized model for each client rather than learning a single global model [2, 30, 26, 46, 6]. FedPer [2] is such a personalized FL method, which shares only the base layers while having local personalized layers for each client, to keep the local knowledge. Unlike the commonly studied image and text data, graph-structured data is defined by connections between instances, and consequently introduces additional challenges: missing edges and shared nodes between private subgraphs. Note that, regarding architectures, there is literature [29, 27, 38, 49] that leverages outputs of neural networks for predicting/minimizing outputs across different client models; however, we use functional outputs of neural networks to identify interconnected subgraphs, thus ours differs from them methodologically.

**Graph Federated Learning**  Few recent studies propose to use the FL framework to collaboratively train GNNs without sharing graph data [13], which can be broadly classified into subgraph- and graph-level methods. Graph-level FL methods assume that different clients have completely disjoint graphs (e.g., molecular graphs), and recent works [44, 14] focus on the heterogeneity among non-IID graphs (i.e., difference in graph labels across various clients). In contrast to graph-level FL methods that have similar challenges to general FL scenarios, the subgraph-level FL problem we target has a unique graph-structural challenge, that there exist missing yet probable links between subgraphs, since a subgraph is a part of a larger global graph. To deal with such a missing link problem among subgraphs, existing methods [42, 45] augment the nodes by requesting the node information in the other subgraphs, and then connecting the existing nodes with the augmented ones. However, this scheme could compromise data privacy constraints, and also increases communication overhead across clients. Unlike existing subgraph FL that focuses on the problem of missing links, our subgraph FL method tackles the problem with a completely different perspective, focusing on discovering subgraph communities [35, 9, 34], which are groups of densely connected subgraphs.

## 3 Personalized Subgraph Federated Learning

We provide the general descriptions of Graph Neural Networks (GNNs) and Federated Learning (FL), and then define our novel problem of personalized subgraph FL lying at the intersection of them.

**Graph Neural Networks**  A graph $\mathcal{G} = (\mathcal{V}, \mathcal{E})$ consists of a set of nodes $\mathcal{V}$ with $n$ elements and a set of edges $\mathcal{E}$ with $m$ elements along with its node feature matrix $\boldsymbol{X} \in \mathbb{R}^{n \times d}$, where each column represents a $d$-dimensional feature for each node. Further, $(u, v) \in \mathcal{E}$ represents an edge from a node $u$ to a node $v$. Then, given the graph, Graph Neural Networks (GNNs) [8, 10] generally represent each node based on features from its neighbors as well as itself, formally defined as follows:

$$\boldsymbol{H}_v^{(l+1)} = \text{UPDATE}^{(l)} \left( \boldsymbol{H}_v^{(l)}, \text{AGGREGATE}^{(l)} \left( \left\{ \boldsymbol{H}_u^{(l)} : \forall u \in \mathcal{N}(v) \right\} \right) \right), \tag{1}$$

where $\boldsymbol{H}_v^{(l)}$ is the feature matrix for node $v$ at $l$-th layer, $\mathcal{N}(v)$ denotes a set of adjacent nodes of node $v$: $\mathcal{N}(v) = \{ u \in \mathcal{V} \mid (u, v) \in \mathcal{E} \}$, AGGREGATE aggregates the features of $v$'s neighbors, and UPDATE updates the node $v$'s representation given its previous representation and the aggregated representations from the neighbors. $\boldsymbol{H}^{(1)}$ is initialized as input node features $\boldsymbol{X}$.

**Federated Learning** The objective of Federated Learning (FL) is to collaboratively train a model with local private data. Let assume that we have $K$ participants with locally collected data that is not accessible from others: $\mathcal{D}_k = \{\boldsymbol{X}_i, \boldsymbol{y}_i\}_{i=1}^{N_k}$, where $\boldsymbol{X}_i$ is a data instance, $\boldsymbol{y}_i$ is its corresponding class label, and $N_k$ is the number of data instances at $k$-th client. Then, for decentralized training with local data, a popular FL algorithm, FedAvg [32], works as the following three steps:

1. **(Initialization)** At the initial communication round $r = 0$, the central server first selects $K$ clients that are available for training, and initializes their local model parameters as the global parameter $\bar{\boldsymbol{\theta}}$, represented as follows: $\boldsymbol{\theta}_k^{(0)} \leftarrow \bar{\boldsymbol{\theta}}^{(0)} \; \forall k$, where $\boldsymbol{\theta}_k^{(0)}$ is the parameters for $k$-th client.

2. **(Local Updates)** Each active local model performs training on private local data $\mathcal{D}_k$ to minimize the task loss $\mathcal{L}(\mathcal{D}_k; \boldsymbol{\theta}_k^{(0)})$, consequently updating the parameters $\boldsymbol{\theta}_k^{(1)} \leftarrow \boldsymbol{\theta}_k^{(0)} - \eta \nabla \mathcal{L}$.

3. **(Global Aggregation)** After local training, the server aggregates the locally learned knowledge with respect to the number of training instances, i.e., $\bar{\boldsymbol{\theta}}^{(1)} \leftarrow \frac{N_k}{N} \sum_{k=1}^{K} \boldsymbol{\theta}_k^{(1)}$ with $N = \sum_k N_k$, and distributes the updated global parameters $\bar{\boldsymbol{\theta}}^{(1)}$ to the local clients selected at the next round.

This FL algorithm iterates between Step 2 and 3 until reaching the final round $R$.

**Challenges in Subgraph FL** While the above FL works well on image and text data, due to the unique structure of graphs, there exist nontrivial challenges for applying this FL scheme to graph-structured data. In particular, unlike with an image domain where each instance $\boldsymbol{X}_i$ is independent from the other images, each node $v$ in a graph is always influenced by its relationships to adjacent nodes $\mathcal{N}(v)$. Moreover, a local graph $G_i$ could be a subgraph of a larger global graph $\mathcal{G}$: $G_i \subseteq \mathcal{G}$. In such a case, there could be missing edges between local subgraphs in two different clients: $(u, v)$ with $u \in \mathcal{V}_i$ and $v \in \mathcal{V}_j$ for clients $i$ and $j$, respectively. To tackle this missing edge problem, few existing subgraph FL methods [42, 45] estimate the nodes from a local subgraph $G_k$ based on the node information from the subgraphs at other clients $G_i \; \forall i \neq k$, and then extend the existing nodes with the estimated ones. However, this augmentation scheme incurs high communication costs as it requires sharing node information across clients, which may also violate data privacy constraints [1].

Yet, there exists another issue that makes subgraph FL even more challenging. Assume that we have a global graph consisting of all the subgraphs. Then, there exists *communities* of such subgraphs [35, 9, 34], where subgraphs within the same community are more densely connected to each other than subgraphs outside the community. Formally, a global graph $\mathcal{G}$ can be decomposed into $T$ different communities: $C_i \subseteq \mathcal{G} \; \forall i = 1, ..., T$, where $i$-th community $C_i = (\mathcal{V}_i, \mathcal{E}_i)$ consists of densely connected nodes. Then, in a subgraph FL problem, each client has a local subgraph $G_j$ that belongs to at least a single community[1]: $C_i = \bigcup_{j=1}^{J} G_j$. Note that, based on the theory of network homophily [33], such connected subgraphs within the same community have similar properties, while subgraphs in two opposite communities are not. Such distributional heterogeneity across communities may lead a naive FL algorithm to collapse incompatible knowledge across different communities.

**Personalized Subgraph FL** To prevent the above knowledge collapse issue, we aim to personalize the subgraph FL algorithm by performing weighted averaging of the local model parameters at the server, rather than learning a single set of global parameters; thereby capturing the subgraph community structures among interrelated subgraphs. Formally, the objective of existing subgraph FL [42, 45, 28] is as follows: $\min_{\boldsymbol{\theta}} \sum_{G_i \subseteq \mathcal{G}} \mathcal{L}(G_i; \boldsymbol{\theta})$. However, a major drawback of such a scheme is that, since the subgraphs in two different communities with sparse connections are extremely heterogeneous due to network homophily [33], finding a universal set of parameters (i.e., $\boldsymbol{\theta}$) that work on all tasks will result in finding a suboptimal parameter set. To address such limitations of existing subgraph FL, we formulate a novel problem of personalized subgraph FL, formalized as follows:

$$\min_{(\boldsymbol{\theta}_i)} \sum_{G_i \subseteq \mathcal{G}} \mathcal{L}(G_i; \boldsymbol{\theta}_i), \; \boldsymbol{\theta}_i \leftarrow \sum_{j=1}^{K} \alpha_{ij} \boldsymbol{\theta}_j \text{ with } \alpha_{ik} \gg \alpha_{il} \text{ for } G_k \subseteq C \text{ and } G_l \nsubseteq C, \quad (2)$$

where $\boldsymbol{\theta}_i$ is the weight for subgraph $G_i$ belonging to community $C$, and $\alpha_{ij}$ is the coefficient for weight aggregation which we will specify in Section 4.1. This formulation promotes the collaborative learning across multiple local models that work on the interrelated subgraphs that belong to the same community, by assigning larger weights on them.

---

[1] For simplicity, we assume that a subgraph belongs to only a single community, however, the formulation does not change even when a subgraph can belong to multiple communities.

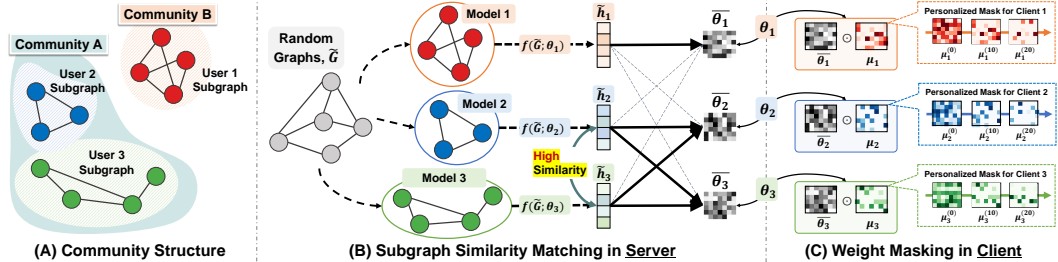

Figure 2: **(A) Two communities**, each of which consists of one/two subgraphs. **(B) Client Similarity Matching**: we forward randomly generated graphs to models $f(\bar{G}; \boldsymbol{\theta})$, and then obtain the functional embeddings of them $\tilde{\boldsymbol{h}}$, which are then used to estimate the similarities between subgraphs. The similarities are used in the weight aggregation, resulting in the personalized model weights $\bar{\boldsymbol{\theta}}$. **(C) Weight Masking:** the transmitted weights from the server to clients $\bar{\boldsymbol{\theta}}$ are masked and shifted by local masks for localization to the local subgraph distribution.

## 4 Federated Personalized Subgraph Learning (FED-PUB) Framework

Our goal of personalized subgraph FL is to jointly improve the local models trained on the inter-connected local subgraphs forming the community structures. To this end, we propose to compute subgraph similarity scores for detecting communities, and to mask subgraph-irrelevant weights.

### 4.1 Subgraph Similarity Estimation for Detecting Subgraph Community

We aim to reflect the community structure consisting of a group of densely connected subgraphs, by sharing more weights among subgraphs in the same community, as formalized in equation 2. Due to network homophily where similar instances in the graph are more associated with each other [33], the subgraphs within the same community should have similar properties. Therefore, if one can measure the subgraph similarities, we can group the similar ones into the community. However, measuring the similarity between local subgraphs is challenging since we do not know which subgraph each client has due to local accessibility. How can we then compute subgraph similarities, without accessing them? To this end, we aim to approximate the subgraph similarity at local clients using auxiliary information obtained from the local GNN models that work on the subgraphs.

**Subgraph Similarity Estimation with Model Parameters** For measuring the similarity between subgraphs at each client, without accessing them, we may use the model parameters as proxies, as follows: $S(i,j) = (\boldsymbol{\theta}_i \cdot \boldsymbol{\theta}_j)/(\|\boldsymbol{\theta}_i\|\|\boldsymbol{\theta}_j\|)$, where $\boldsymbol{\theta}$ is a flattened parameter into the vector, and $S$ is a similarity measure. This may sound reasonable since the GNN model trained on the subgraph will embed its knowledge into its parameters. However, this scheme has a notable drawback that similarity measured in the high-dimensional parameter space is not meaningful due to the curse of dimensionality [4], and that the cost of calculating the similarity between parameters grows rapidly as the model size increases (See Figure 3).

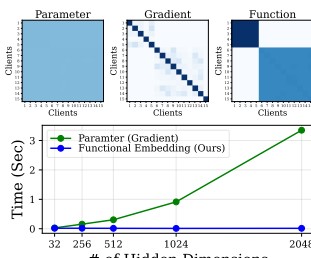

Figure 3: Effectiveness (top) and efficiencies (bottom) of different similarity measurements.

**Subgraph Similarity Estimation with Functional Embedding** To tackle the limitations of using parameter distance, we propose to measure the functional similarity of neural networks by feeding the same input to every local client and then calculating the similarities using their outputs, inspired by a work for neural network search [17]. The main intuition is that we can consider the transformation defined with a neural network as a function, and we measure the functional similarity of two networks by the distance of their outputs for the same input. However, unlike the previous work [17] that tackles image classification, which uses Gaussian noises as inputs, we use random graphs as inputs as we work with GNNs. Formally, let $\tilde{G} = (\tilde{\mathcal{V}}, \tilde{\mathcal{E}})$ be a random community graph obtained from a stochastic block model [15], where subgraphs within the community have more edges between them than edges across the communities. Further, $\tilde{\mathcal{V}}$ is randomly initialized from the normal distribution. Then, the similarity between two functions defined by GNNs at clients $i$ and $j$ is defined as follows:

$$S(i,j) = \frac{\tilde{\boldsymbol{h}}_i \cdot \tilde{\boldsymbol{h}}_j}{\|\tilde{\boldsymbol{h}}_i\|\|\tilde{\boldsymbol{h}}_j\|}, \quad \tilde{\boldsymbol{h}}_i = \texttt{AVG}(f(\tilde{G}; \boldsymbol{\theta}_i)) \text{ and } \tilde{\boldsymbol{h}}_j = \texttt{AVG}(f(\tilde{G}; \boldsymbol{\theta}_j)), \tag{3}$$

where $\tilde{\boldsymbol{h}}$ is the averaged output of all node embeddings for input $\tilde{G}$ with $\texttt{AVG}$ operation to reduce the dimensionality of the output from $n \times d$ to $d$, for $n$ nodes with $d$-dimensional node features.

**Personalized Weight Aggregation based on Subgraph Similarity**  With equation 3, the remaining
step is then to share the model weights between models working on similar subgraphs belonging to
the same community. However, entirely ignoring the model parameters from different communities
may result in exploiting only the local objective while ignoring globally useful weights, which may
result in performance degeneration. Therefore, we perform weighted averaging of all the local models
from the other clients based on their functional (subgraph) similarities, as follows (Figure 2 (B)):

$$\bar{\boldsymbol{\theta}}_i \leftarrow \sum_j \alpha_{ij} \cdot \boldsymbol{\theta}_j, \quad \alpha_{ij} = \frac{\exp(\tau \cdot S(i,j))}{\sum_k \exp(\tau \cdot S(i,k))}, \tag{4}$$

where $\alpha_{ij}$ is a normalized similarity between clients $i$ and $j$, and $\tau$ is a hyperparameter for scaling
the unnormalized similarity score. Note that increasing the value of $\tau$ (e.g., 10) will result in model
averaging done almost exclusively among subgraphs detected as belonging to the same community.

This personalized scheme handles two challenges in subgraph FL. First, in contrast to the global
weight aggregation scheme which easily collapses the knowledge from heterogeneous communities
into a single model, our subgraph FL allows the models belonging to different communities to obtain
model weights that are beneficial for each community. Also, the missing edges between subgraphs
that have been explicitly handled by previous works [42, 45] could be also implicitly considered by
assigning larger weights to models within the same community (See Figure 10). This also enhances
data privacy while minimizing the communication costs between probably linked subgraphs.

### 4.2  Adaptive Weight Masking for Selecting Subgraph-Relevant Parameters

With the previous similarity matching scheme, we can effectively group GNN models that belong to
the same community, thus preventing the collapsing of irrelevant knowledge from other communities.
However, the scalar weighting scheme only considers how much each local model from other clients
is relevant for the subgraph task, but not *which* parameters are relevant. Thus we propose a scheme to
select only the relevant parameters from the aggregated model weights transmitted from the server.

**Personalized Parameter Masking**  We perform selective training and updating of the aggregated
parameters by modulating and shifting them, using sparse local masks. Formally, let $\boldsymbol{\mu}_k$ be a local
mask for a client $k$. Then, our local model weight is obtained by modulating the weights from the
server, as follows: $\boldsymbol{\theta}_k = \bar{\boldsymbol{\theta}}_k \odot \boldsymbol{\mu}_k$, where $\odot$ is an element-wise multiplication operation between
the globally given weight $\bar{\boldsymbol{\theta}}_k$ and the local mask $\boldsymbol{\mu}_k$. Note that the local mask is a free variable
and is not shared across clients. Also, we initialize $\boldsymbol{\mu}_k$ as ones, in order to start training with the
globally initialized model parameters without modification. We then further promote sparsity on
the mask, which brings two key advantages. First, we can transmit only the partial parameters, that
have not been sparsified at the client to the server rather than sending all parameters, thus reducing
the communication costs. Moreover, if local masks are sufficiently sparse, the local models can be
trained faster, given that zero-skipping operations are supported (Figure 2 (C)). To take these benefits
in sparsity, we use $L_1$ regularizer on $\boldsymbol{\mu}_k$ when performing local optimization, as shown in equation 5.

**Preventing Local Divergence with Proximal Term**  As masks are trained only with limited local
data without parameter sharing, they may be easily overfitted to the training instances in each client.
To alleviate this issue, we adopt the proximal term proposed in Li et al. [25] that regularizes the locally
updated models $\boldsymbol{\theta}_k$ to be closer to the globally given model $\bar{\boldsymbol{\theta}}_k$, therefore, preventing the model from
extremely drifting to the local training distribution. To sum up, at $k$-th client, our objective function
including sparsity and proximal terms with $L_1$ and $L_2$ losses is denoted as follows:

$$\min_{(\boldsymbol{\theta}_k, \boldsymbol{\mu}_k)} \mathcal{L}(\mathcal{D}_k; \boldsymbol{\theta}_k, \boldsymbol{\mu}_k) + \lambda_1 \|\boldsymbol{\mu}_k\|_1 + \lambda_2 \|\boldsymbol{\theta}_k - \bar{\boldsymbol{\theta}}_k\|_2^2, \tag{5}$$

where $\mathcal{L}$ is the conventional cross-entropy loss function, and $\lambda_1$ and $\lambda_2$ are scaling hyper-parameters.

## 5  Experiments

We now experimentally validate our FED-PUB on six different datasets under both the overlapping
and disjoint subgraph scenarios with varying client numbers, with node classification tasks.

### 5.1  Experimental Setups

**Datasets**  Following the setup from Zhang et al. [45], we construct the distributed subgraphs from
the benchmark dataset by dividing it into the number of participants, each of which has a subgraph
that is a part of an original graph. Specifically, we use six datasets: Cora, CiteSeer, Pubmed and

Table 1: **Results on the overlapping node scenario.** The reported results are mean and standard deviation over three different runs. Only the statistically significant performances ($p > 0.05$) are highlighted in bold.

| | Cora | | | CiteSeer | | | Pubmed | | | - |
|---|---|---|---|---|---|---|---|---|---|---|
| Methods | 10 Clients | 30 Clients | 50 Clients | 10 Clients | 30 Clients | 50 Clients | 10 Clients | 30 Clients | 50 Clients | - |
| Local | 73.98 ± 0.25 | 71.65 ± 0.12 | 76.63 ± 0.10 | 65.12 ± 0.08 | 64.54 ± 0.42 | 66.68 ± 0.44 | 82.32 ± 0.07 | 80.72 ± 0.16 | 80.54 ± 0.11 | - |
| FedAvg | 76.48 ± 0.36 | 53.99 ± 0.98 | 53.99 ± 4.53 | 69.48 ± 0.15 | 66.15 ± 0.64 | 66.51 ± 1.00 | 82.67 ± 0.11 | 82.05 ± 0.12 | 80.24 ± 0.35 | - |
| FedProx | 77.85 ± 0.50 | 51.38 ± 1.74 | 56.27 ± 9.04 | 69.39 ± 0.35 | 66.11 ± 0.75 | 66.53 ± 0.43 | 82.63 ± 0.17 | 82.13 ± 0.13 | 80.50 ± 0.46 | - |
| FedPer | 78.73 ± 0.31 | 74.18 ± 0.24 | 74.42 ± 0.37 | 69.81 ± 0.28 | 65.19 ± 0.81 | 67.64 ± 0.44 | 85.31 ± 0.06 | 84.35 ± 0.38 | 83.94 ± 0.10 | - |
| GCFL | 78.84 ± 0.26 | 73.41 ± 0.27 | 76.63 ± 0.16 | 69.48 ± 0.39 | 64.92 ± 0.18 | 65.98 ± 0.30 | 83.59 ± 0.25 | 80.77 ± 0.12 | 81.36 ± 0.11 | - |
| FedGNN | 70.63 ± 0.83 | 61.38 ± 2.33 | 56.91 ± 0.82 | 68.72 ± 0.39 | 59.98 ± 1.52 | 58.98 ± 0.98 | 84.25 ± 0.07 | 82.02 ± 0.22 | 81.85 ± 0.19 | - |
| FedSage+ | 77.52 ± 0.46 | 51.99 ± 0.42 | 55.48 ± 11.5 | 68.75 ± 0.48 | 65.97 ± 0.02 | 65.93 ± 0.30 | 82.77 ± 0.08 | 82.14 ± 0.11 | 80.31 ± 0.68 | - |
| FED-PUB (Ours) | **79.60 ± 0.12** | **75.40 ± 0.54** | **77.84 ± 0.23** | **70.58 ± 0.20** | **68.33 ± 0.45** | **69.21 ± 0.30** | **85.70 ± 0.08** | **85.16 ± 0.10** | **84.84 ± 0.12** | - |

| | Amazon-Computer | | | Amazon-Photo | | | ogbn-arxiv | | | All |
|---|---|---|---|---|---|---|---|---|---|---|
| Methods | 10 Clients | 30 Clients | 50 Clients | 10 Clients | 30 Clients | 50 Clients | 10 Clients | 30 Clients | 50 Clients | Avg. |
| Local | 88.50 ± 0.20 | 86.66 ± 0.00 | 87.04 ± 0.02 | 92.17 ± 0.12 | 90.16 ± 0.12 | 90.42 ± 0.15 | 62.52 ± 0.07 | 61.32 ± 0.04 | 60.04 ± 0.04 | 76.72 |
| FedAvg | 88.99 ± 0.19 | 83.37 ± 0.47 | 76.34 ± 0.12 | 92.91 ± 0.07 | 89.30 ± 0.22 | 74.19 ± 0.57 | 63.56 ± 0.02 | 59.72 ± 0.06 | 60.94 ± 0.24 | 73.38 |
| FedProx | 88.84 ± 0.20 | 83.84 ± 0.89 | 76.60 ± 0.47 | 92.67 ± 0.19 | 89.17 ± 0.40 | 72.36 ± 2.06 | 63.52 ± 0.11 | 59.86 ± 0.16 | 61.12 ± 0.04 | 73.38 |
| FedPer | 89.30 ± 0.04 | 87.99 ± 0.23 | 88.22 ± 0.27 | 92.88 ± 0.24 | 91.23 ± 0.16 | 90.92 ± 0.38 | 63.97 ± 0.08 | 62.29 ± 0.04 | 61.24 ± 0.11 | 78.42 |
| GCFL | 89.01 ± 0.22 | 87.24 ± 0.09 | 87.02 ± 0.22 | 92.45 ± 0.10 | 90.58 ± 0.11 | 90.54 ± 0.08 | 63.24 ± 0.07 | 61.66 ± 0.10 | 60.32 ± 0.01 | 77.61 |
| FedGNN | 88.15 ± 0.09 | 87.00 ± 0.10 | 83.96 ± 0.88 | 91.47 ± 0.11 | 87.91 ± 1.34 | 78.90 ± 6.46 | 63.08 ± 0.19 | 60.09 ± 0.04 | 60.51 ± 0.11 | 73.66 |
| FedSage+ | 89.24 ± 0.15 | 81.33 ± 1.20 | 76.72 ± 0.39 | 92.76 ± 0.05 | 88.69 ± 0.99 | 72.41 ± 1.36 | 63.24 ± 0.02 | 59.90 ± 0.12 | 60.95 ± 0.09 | 73.12 |
| FED-PUB (Ours) | **89.98 ± 0.08** | **89.15 ± 0.06** | **88.76 ± 0.14** | **93.22 ± 0.07** | **92.01 ± 0.07** | **91.71 ± 0.11** | **64.18 ± 0.04** | **63.34 ± 0.12** | **62.55 ± 0.12** | **79.53** |

Figure 4: **Convergence plots for the overlapping node scenario.** We visualize the test accuracy curves for all six datasets corresponding to Table 1, over 100 communication rounds with 10 clients.

ogbn-arxiv for citation graphs [39, 16]; Computer and Photo for product graphs [31, 40]. We then divide the original graph into multiple subgraphs using the METIS graph partitioning algorithm [20]. Note that, unlike the Louvain algorithm [5] presented in Zhang et al. [45] that requires to further merge partitioned subgraphs into particular numbers of subgraphs since it cannot specify the number of subsets (i.e., clients for FL), the METIS algorithm can specify the number of subsets, thus making more reasonable experimental settings in subgraph FL (See Section C.2 of the supplementary file). For the non-overlapping scenario where there are no duplicate nodes between subgraphs, we use the output from the METIS as it provides the non-overlapping partitions. Meanwhile, for the overlapping scenario where nodes are duplicated among subgraphs, we randomly sample the subgraphs multiple times from the partitioned graph. For more details, please see Section B of the supplementary file.

**Baselines**   1) **FedAvg** [32] and 2) **FedProx** [25]: The most popular FL baselines. 3) **FedPer** [2]: A personalized FL baseline without sharing personalized layers. 4) **FedGNN** [42] and 5) **FedSage+** [45]: Subgraph FL baselines which we mainly target. 6) **GCFL** [44]: A graph FL baseline which learns completely disjoint graphs as in clustered FL [37], adopted for subgraph FL. 7) **Local**: A baseline without sharing weights with other clients. 8) **FED-PUB**: Our personalized subgraph FL including subgraph similarity matching and weight masking. See Section B of the supplementary file for details.

**Implementation Details**   We set the GCN [22] with two layers as the base GNN for all models. We perform federated learning over 100 communication rounds for Cora, CiteSeer and Pubmed datasets, while 200 rounds for Computer, Photo and arxiv datasets, considering the size of datasets. The local training epoch is selected in the range of $\{1, 2, 3\}$ depending on the dataset size (e.g., Computer is three while CiteSeer is one)[2]. We use the Adam optimizer [21] for model optimization. We then measure the node classification accuracy on subgraphs at the client-side, and then average the performance across clients. We provide further details in Section B of the supplementary file.

## 5.2   Experimental Results

**Main Results**   Table 1 shows the node classification performance under the overlapping subgraph scenario, in which our FED-PUB statistically ($p > 0.05$) significantly outperforms all the baselines. In particular, while FedGNN and FedSage+ are two pioneer works for the subgraph FL problem, they significantly underperform personalized FL methods including ours, especially at the larger number of clients. This is even surprising as they share node information between clients for handling the missing edge problem, yet we suppose such inferior performance comes from naive averaging of local weights without consideration of community structures. While personalized FL baselines including FedPer and GCFL show decent performance by alleviating the knowledge collapse between subgraphs with local parameters or clustering, they still largely underperform ours as they are not concerned with the aggregation between similar subgraphs that form a community (i.e., GCFL uses a bi-partitioning scheme where it iteratively divides a group of subgraphs within the same community

---

[2]We found communication rounds and local epochs are important factors to prevent overfitting of all models.

Table 2: **Results on the non-overlapping node scenario.** The reported results are mean and standard deviation over three different runs. Only the statistically significant performances ($p > 0.05$) are highlighted in bold.

| Methods | Cora | | | CiteSeer | | | Pubmed | | | - |
|---|---|---|---|---|---|---|---|---|---|---|
| | 5 Clients | 10 Clients | 20 Clients | 5 Clients | 10 Clients | 20 Clients | 5 Clients | 10 Clients | 20 Clients | - |
| Local | $81.30 \pm 0.21$ | $79.94 \pm 0.24$ | $80.30 \pm 0.25$ | $69.02 \pm 0.05$ | $67.82 \pm 0.13$ | $65.98 \pm 0.17$ | $84.04 \pm 0.18$ | $82.81 \pm 0.39$ | $82.65 \pm 0.03$ | - |
| FedAvg | $74.45 \pm 5.64$ | $69.19 \pm 0.67$ | $69.50 \pm 3.58$ | $71.06 \pm 0.60$ | $63.61 \pm 3.59$ | $64.68 \pm 1.83$ | $79.40 \pm 0.11$ | $82.71 \pm 0.29$ | $80.97 \pm 0.26$ | - |
| FedProx | $72.03 \pm 4.56$ | $60.18 \pm 7.04$ | $48.22 \pm 6.81$ | $71.73 \pm 1.11$ | $63.33 \pm 3.25$ | $64.85 \pm 1.35$ | $79.45 \pm 0.25$ | $82.55 \pm 0.24$ | $80.50 \pm 0.25$ | - |
| FedPer | $81.68 \pm 0.40$ | $79.35 \pm 0.04$ | $78.01 \pm 0.32$ | $70.41 \pm 0.32$ | $70.53 \pm 0.28$ | $66.64 \pm 0.27$ | $85.80 \pm 0.21$ | $84.20 \pm 0.28$ | $84.72 \pm 0.31$ | - |
| GCFL | $81.47 \pm 0.65$ | $78.66 \pm 0.27$ | $79.21 \pm 0.70$ | $70.34 \pm 0.57$ | $69.01 \pm 0.12$ | $66.33 \pm 0.05$ | $85.14 \pm 0.33$ | $84.18 \pm 0.19$ | $83.94 \pm 0.36$ | - |
| FedGNN | $81.51 \pm 0.68$ | $70.12 \pm 0.99$ | $70.10 \pm 3.52$ | $69.06 \pm 0.92$ | $55.52 \pm 3.17$ | $52.23 \pm 6.00$ | $79.52 \pm 0.23$ | $83.25 \pm 0.45$ | $81.61 \pm 0.59$ | - |
| FedSage+ | $72.97 \pm 5.94$ | $69.05 \pm 1.59$ | $57.97 \pm 12.6$ | $70.74 \pm 0.69$ | $65.63 \pm 3.10$ | $65.46 \pm 0.74$ | $79.57 \pm 0.24$ | $82.62 \pm 0.31$ | $80.82 \pm 0.25$ | - |
| FED-PUB (Ours) | $\mathbf{83.70 \pm 0.19}$ | $\mathbf{81.54 \pm 0.12}$ | $\mathbf{81.75 \pm 0.56}$ | $\mathbf{72.68 \pm 0.44}$ | $\mathbf{72.35 \pm 0.53}$ | $\mathbf{67.62 \pm 0.12}$ | $\mathbf{86.79 \pm 0.09}$ | $\mathbf{86.28 \pm 0.18}$ | $\mathbf{85.53 \pm 0.30}$ | - |

| Methods | Amazon-Computer | | | Amazon-Photo | | | ogbn-arxiv | | | All |
|---|---|---|---|---|---|---|---|---|---|---|
| | 5 Clients | 10 Clients | 20 Clients | 5 Clients | 10 Clients | 20 Clients | 5 Clients | 10 Clients | 20 Clients | Avg. |
| Local | $89.22 \pm 0.13$ | $88.91 \pm 0.17$ | $89.52 \pm 0.20$ | $91.67 \pm 0.09$ | $91.80 \pm 0.02$ | $90.47 \pm 0.15$ | $66.76 \pm 0.07$ | $64.92 \pm 0.09$ | $65.06 \pm 0.05$ | 79.57 |
| FedAvg | $84.88 \pm 1.96$ | $79.54 \pm 0.23$ | $74.79 \pm 0.24$ | $89.89 \pm 0.83$ | $83.15 \pm 3.71$ | $81.35 \pm 1.04$ | $65.54 \pm 0.07$ | $64.44 \pm 0.10$ | $63.24 \pm 0.13$ | 74.58 |
| FedProx | $85.25 \pm 1.27$ | $83.81 \pm 1.09$ | $73.05 \pm 1.30$ | $90.38 \pm 0.48$ | $80.92 \pm 4.64$ | $82.32 \pm 0.29$ | $65.21 \pm 0.20$ | $64.37 \pm 0.18$ | $63.03 \pm 0.04$ | 72.84 |
| FedPer | $89.67 \pm 0.34$ | $89.73 \pm 0.04$ | $87.86 \pm 0.43$ | $91.44 \pm 0.37$ | $91.76 \pm 0.23$ | $90.59 \pm 0.06$ | $66.87 \pm 0.05$ | $64.99 \pm 0.18$ | $64.66 \pm 0.11$ | 79.94 |
| GCFL | $89.07 \pm 0.91$ | $90.03 \pm 0.16$ | $89.08 \pm 0.25$ | $91.99 \pm 0.29$ | $92.06 \pm 0.25$ | $90.79 \pm 0.17$ | $66.80 \pm 0.12$ | $65.09 \pm 0.08$ | $65.08 \pm 0.04$ | 79.90 |
| FedGNN | $88.08 \pm 0.15$ | $88.18 \pm 0.41$ | $83.16 \pm 0.13$ | $90.25 \pm 0.70$ | $87.12 \pm 2.01$ | $81.00 \pm 4.48$ | $65.47 \pm 0.22$ | $64.21 \pm 0.32$ | $63.80 \pm 0.05$ | 75.23 |
| FedSage+ | $85.04 \pm 0.61$ | $80.50 \pm 1.30$ | $70.42 \pm 0.85$ | $90.77 \pm 0.44$ | $76.81 \pm 8.24$ | $80.58 \pm 1.15$ | $65.69 \pm 0.09$ | $64.52 \pm 0.14$ | $63.31 \pm 0.20$ | 73.47 |
| FED-PUB (Ours) | $\mathbf{90.74 \pm 0.05}$ | $\mathbf{90.55 \pm 0.13}$ | $\mathbf{90.12 \pm 0.09}$ | $\mathbf{93.29 \pm 0.19}$ | $\mathbf{92.73 \pm 0.18}$ | $\mathbf{91.92 \pm 0.12}$ | $\mathbf{67.77 \pm 0.09}$ | $\mathbf{66.58 \pm 0.08}$ | $\mathbf{66.64 \pm 0.12}$ | **81.59** |

| (a) Cora | (b) CiteSeer | (c) Pubmed | (d) Computer | (e) Photo | (f) ogbn-arxiv |
|---|---|---|---|---|---|

Figure 5: **Convergence plots for the non-overlapping node scenario.** We visualize the test accuracy curves for all six datasets corresponding to Table 2, over 100 communication rounds with 10 clients.

into two disjoint sets). We then further conduct the experiments on the disjoint subgraph scenarios (non-overlapping scenario), where nodes are not overlapped between subgraphs, which makes the subgraph FL problem more heterogeneous. As shown in Table 2, FED-PUB consistently outperforms all existing baselines in such a challenging scenario, demonstrating the efficacy of ours.

**Fast Local Convergence**    As shown in Figure 4 and 5, our FED-PUB converges rapidly, compared against baselines including personalized FL models. We conjecture that this is because, not only ours accurately identifies subgraphs forming the community and then shares weights largely across them for promoting the joint improvement of them, but also masking subgraph-irrelevant weights received from the server for localization to local subgraphs, demonstrated in the next two paragraphs.

**Accurate Community Detection**    We aim to show whether FED-PUB accurately groups subgraphs comprising a community during weight aggregation. If two different subgraphs have many missing edges or have similar label distributions, we usually regard those two as within the same community [35, 9, 34]. Thereby, as shown in Figure 6 (a) and (b), there are four different communities by the interval of five, and the last two communities further comprise a larger community. Then, as shown in Figure 6 (c) and (d), FED-PUB detects obvious four communities at the first few rounds, and then captures the larger yet somewhat less-obvious community consisting of two smaller communities.

**Ablation Study**    To analyze the contribution of each component, we conduct the ablation studies. As shown in Figure 7, we observe that each of our subgraph similarity matching and weight masking significantly improves the performances from the naive FedAvg, while the performance is much improved when using both together. However, the benefit from each component is different across overlapping and non-overlapping scenarios. In particular, in the former scenario where a group of highly overlapped subgraphs usually comprise a community, similarity matching for community detection is more beneficial since capturing the community would promote the joint improvement of subgraphs belonging to the same community. However, in the non-overlapping scenarios, subgraphs within the same community become lesser similar, thus selectively using the aggregated model weights from the server with personalized weight masks improves the performance a lot.

**Communication Efficiency**    Another notable advantage of using the sparse masks is that we can reduce the communication costs at every FL round, as well as the model size for faster training, which we demonstrate in Table 8. In particular, Table 8 shows that existing subgraph FL methods require more than two times larger communications costs, measured by adding both the client-to-server and server-to-client costs, compared against the naive FedAvg, since they require to transfer additional node information between clients for estimating the probable nodes on the subgraphs. Contrarily, our FED-PUB has significantly lower communication costs and lower model sizes by using the sparse masks on the model weights: transmitting and training with only the partial parameters not sparsified at the client. Further, as shown in ours variants in Table 8, we can manage the trade-off between the model sparsity and the performance by controlling the hyperparameter for sparsity regularization, $\lambda_1$.

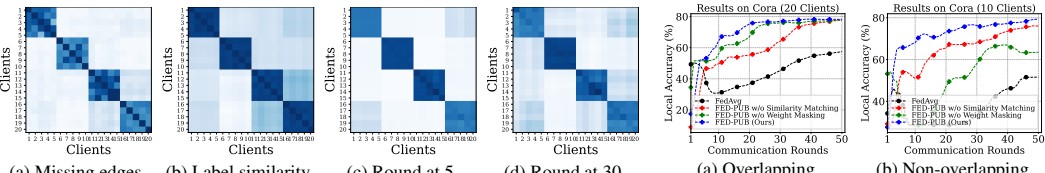

| (a) Missing edges | (b) Label similarity | (c) Round at 5 | (d) Round at 30 | (a) Overlapping | (b) Non-overlapping |

Figure 6: **The heatmaps of the community structure** on the overlapping node scenario with Cora (20 clients). Dark color indicates lots of missing edges between subgraphs (a) or high similarities in labels (b). (c) and (d) are functional similarities captured by our FED-PUB.

Figure 7: **Ablation studies** of our FED-PUB on both the overlapping (a) and non-overlapping (b) subgraph scenarios, on the Cora dataset.

| Model | Acc. [%] | Model Size [%] | Cost [%] |
|---|---|---|---|
| FedAvg | 76.48 ± 0.36 | 100.00 ± 0.00 | 100.00 ± 0.00 |
| FedGNN | 70.63 ± 0.83 | 100.00 ± 0.00 | 214.94 ± 0.00 |
| FedSage+ | 77.52 ± 0.46 | 100.00 ± 0.00 | 276.84 ± 0.00 |
| GCFL | 78.84 ± 0.26 | 100.00 ± 0.00 | 100.00 ± 0.00 |
| **Ours** ($\lambda_1$=9e-1) | 77.36 ± 0.99 | **25.13** ± 0.34 | **37.70** ± 0.56 |
| **Ours** ($\lambda_1$=7e-1) | 79.46 ± 0.41 | 42.59 ± 1.33 | 63.89 ± 1.99 |
| **Ours** ($\lambda_1$=5e-1) | **79.89** ± 0.12 | 57.07 ± 0.52 | 85.61 ± 0.78 |

Figure 8: **Analysis on efficiencies** of communication costs and model sizes.

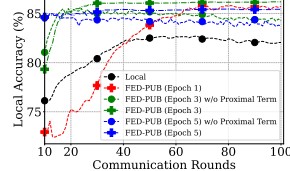

Figure 9: **Varying the local epochs** with accuracy curves.

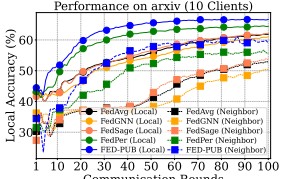

Figure 10: **Performance on neighboring subgraphs**.

**Varying Local Epochs** As shown in Figure 9, when we increase the number of communication rounds and the local steps, the model diverges to the local subgraphs (i.e., overfitting), due to the small number of training instances and the direct connection between training and test nodes: struggle to generalize to the test instances. However, our model with the proximal term in equation 5 alleviates this issue, therefore, maintaining the highest local performance. Notably, the performance with five local epochs is inferior to the performance of one epoch, which indicates that increasing the local epochs does not always bring advantages and properly tuning them is important for subgraph FL.

**Handling Missing Edges** To measure whether FED-PUB can handle the missing edge problem: information is not shared between two neighboring subgraphs due to the missing edges, we use the local model trained on the local subgraph for evaluating the performance on its neighboring subgraph, in which the local subgraph has the most missing edges to its neighboring subgraph. Specifically, in Figure 10, (Neighbor) denotes the subgraph performance evaluated by its neighbor model, while (Local) denotes the subgraph performance from its own local model. Then, the high performance on (Neighbor) measure means two associated subgraphs share meaningful knowledge without having explicit edges between them, thereby solving the missing edge problem. Note that, existing subgraph FL explicitly augments the nodes and edges for capturing the potential information flow over the missing edges between subgraphs, while ours implicitly shares weights a lot across similar subgraphs within the same community. Figure 10 shows that ours achieves the significantly superior performance on the neighboring subgraph problem against subgraph FL baselines, which confirms that ours has an advantage on the missing edge problem by meaningfully sharing knowledge between two subgraphs having potentially missing edges, without explicitly estimating them.

## 6 Conclusion

We introduced a novel problem of personalized subgraph FL, which focuses on the joint improvement of local GNNs working on interrelated subgraphs (e.g. subgraphs belonging to the same community), by selectively utilizing knowledge from other models. The proposed personalized subgraph FL is highly challenging due to 1) difficulty of computing similarities between local subgraphs that are only locally accessible, and 2) knowledge collapse among local models that work on heterogeneous subgraphs during weight aggregation. To this end, we proposed a novel personalized subgraph FL framework, referred to as FEDerated Personalized sUBgraph learning (FED-PUB), which computes the similarities across subgraphs using functional embeddings of their local GNNs on random graphs, and uses them to perform a weighted average of the local models for each client. Further, we mask out globally given weights to focus on only the relevant subnetwork for each client (or community). We extensively validated our framework on multiple benchmark datasets with both overlapping and non-overlapping subgraphs, on which our FED-PUB significantly outperforms relevant baselines. Further analyses show the effectiveness of the subgraph similarity matching for detecting the community structures, as well as the weight masking for tackling the subgraph heterogeneity. We provide the limitations and potential societal impacts of our work in Section D of the supplementary file.

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
