# OpenReview forum: "Personalized Subgraph Federated Learning"
_NeurIPS.cc/2022/Conference — NeurIPS 2022 Submitted_

### Official Review · Reviewer_gnhR · 2022-07-09

**Rating:** 5
**Confidence:** 4
**Soundness:** 2 fair
**Presentation:** 3 good
**Contribution:** 2 fair

**Summary:**

Aiming to solve the data heterogeneity problem in subgraph federated learning (FL), this paper presents a novel research problem, namely personalised subgraph FL. Focus on this personalised FL scenario, the authors propose a new learning method named FED-PUB. To train local models effectively, FED-PUB considers two key designs, i.e., functional embedding-based similarity estimation and adaptive parameter masking. Extensive experiments validate the effectiveness of FED-PUB.

**Questions:**

1. What is the meaning of J in Eq. (2)? If J is the number of subgraphs within a community, how to find a specific community for each subgraph during model training? And does that mean the aggregation is only executed within the subgraph in a community?
2. In Line 45, the authors claim that the algorithms in [40] and [43] may compromise data privacy. Is there any further evidence to prove that?
3. In the formulation of existing subgraph FL (Line 169) and personalized FL (Eq. (2)), what is the optimized parameter? It's better to highlight this under the "min" operation.
4. Could you provide a theoretical analysis for the techniques proposed in this paper?

**Limitations:**

The authors have discussed the limitations and potential negative social impacts in Appendix D.

**Strengths And Weaknesses:**

Pros:
1. The research problem is interesting.
This paper first proposes to study the personalized problem of graph federated learning. The novel research problem is interesting and challenging, which may inspire future studies in the community.
2. The experiments are extensive.
The authors conduct sufficient experiments to compare the performance with baseline methods and discover the property of the proposed method. The proposed method achieves outstanding results, which proves its effectiveness.

Cons:
1. The novelty of the proposed method is limited.
Despite some interesting designs in the proposed model, the core idea of this paper is not novel enough and similar to some existing works. For instance, using shared data to calculate the similarity between local models has been proposed in [*1]. There is a branch of works that incorporate auxiliary data to assist the federated training procedure, such as [*2], [*3], and [*4].
[*1] Makhija, Disha, et al. "Architecture Agnostic Federated Learning for Neural Networks." arXiv preprint arXiv:2202.07757 (2022). (Accepted by ICML 2022)
[*2] Lin, Tao, et al. "Ensemble distillation for robust model fusion in federated learning." Advances in Neural Information Processing Systems 33 (2020): 2351-2363.
[*3] Sattler, Felix, et al. "Fedaux: Leveraging unlabeled auxiliary data in federated learning." IEEE Transactions on Neural Networks and Learning Systems (2021).
[*4] Zhu, Zhuangdi, Junyuan Hong, and Jiayu Zhou. "Data-free knowledge distillation for heterogeneous federated learning." International Conference on Machine Learning. PMLR, 2021.
2. The relationship between the key design (i.e. weight masking) and the research problem is unclear.
The authors claim that the weight masking mechanism is to select the relevant parameters for each subgraph. However, it seems like a general solution for personalized FL under heterogeneity and is less relevant to the specific subgraph FL problem. Although it brings some performance improvement, how it benefits the node classification problem is unknown.
3. The motivation behind personalized subgraph FL is not convictive enough.
In the introduction section, the authors use an intuitive explanation (with Fig 1) and a tiny experiment on the synthetic dataset to demonstrate the heterogeneity problem and further show the need for personalized subgraph FL. However, how serious the heterogeneity problem on real-world datasets under the subgraph FL setting is still unknown. The real-world experiments, to a certain extent, demonstrate the issue, but what if the clients' number is much smaller than 10? Maybe more comprehensive experiments will better show the significance of personalized FL.
4. The capability of dealing with missing edges is over-claimed.
In Sec 4.1 (Line 230) and Sec 5.2 (Line 348), the authors claim that FED-PUB can handle the missing edges problem in an implicit manner. This is not persuasive enough. In Fig 10, we can still find that FED-PUB(neighbor) has sub-optimal performance, and the performance gap between (local) and (neighbor) is similar among different methods. In another word, the good performance in this setting cannot prove that the performance gain is caused by handling the missing edges problem.

---

> ### Author Response · Authors · 2022-07-31
> **Initial Response (5/5) to Reviewer gnhR**
>
> **Question 7:** In the formulation of existing subgraph FL (Line 169) and personalized subgraph FL (Eq. (2)), what is the optimized parameter? It's better to highlight this under the "min" operation.
>
> **Answer:** Thank you for pointing it out, and we apologize for the confusion. The optimized parameter is $\theta$ for the existing subgraph FL, and $\theta_i$ for our personalized subgraph FL. Also, after introducing the weight masking scheme in Section 4.2, we additionally optimize the local mask $\mu_i$ along with existing $\theta_i$, as represented in Equation (5).
>
> ---
>
> **Question 8:** Could you provide a theoretical analysis?
>
> **Answer:** Thank you for your comment, and we already motivated our personalized subgraph FL task with theoretical concepts of network homophily and network community in Section 3. More specifically, at first, from the definition of network homophily [31], nodes with similar properties are more likely to connect to each other than dissimilar ones. Then, based on this definition, similar nodes are densely connected, which then form a community from the definition of the community [33, 9, 32] (i.e., the number of edges within the same community is larger than the number of edges estimated by the random model). In this regard, for the subgraph FL problem that we target, we specify two facts from the graph theories above that it is important to capture the community structures to promote joint improvements among subgraphs having similar properties, and to handle the incompatible knowledge issues between subgraphs in different communities. Therefore, to summarize, theoretical concepts of network homophily and network community motivate us to find the challenges on subgraph FL, and, from this, we define our novel personalized subgraph FL problem represented in Equation (2). In other words, we clearly introduce our ideas based on graph theories, although we do not believe that any rigorous proofs are required when introducing them. Furthermore, regarding the methodological perspective, for deep networks, it is challenging to analyze the functional behavior of every deep layer, and then invent a new theoretical justification. For this reason, lots of deep FL works published recently (e.g., [2, 40]) do not provide theoretical analyses, while they are well received due to their task-level ideas and sufficient empirical verifications. Thus, we believe that analyzing the theoretical behavior of our deep FL methods goes beyond the scope of our work, and leave it as future work.

---

> ### Author Response · Authors · 2022-07-31
> **Initial Response (4/5) to Reviewer gnhR**
>
> **Question 4:** The capability of dealing with missing edges is over-claimed. Specifically, in Sec 4.1 (Line 230) and Sec 5.2 (Line 348), the authors claim that FED-PUB can handle the problem of missing edges in an implicit manner with experimental results in Figure 10, which is, however, not persuasive enough since FED-PUB (neighbor) has the sub-optimal performance in Figure 10.
>
> **Answer:** This is a misinterpretation of the results in Figure 10, since our FED-PUB (Local) and (Neighbor) models **perform the best** among all the other (Local) and (Neighbor) models, respectively, which is sufficient evidence that it effectively alleviates the missing edge problem (See the next paragraph for details). Moreover, the qualitative results in Figure 6 also support the missing edge claim. In particular, since our FED-PUB puts high similarities on subgraphs with a large number of missing edges as shown in Figure 6 (a) and (d), it can inherently handle the missing edge problem: it enables to share knowledge across potentially connected, yet disconnected subgraphs with missing edges.
>
> Specifically, regarding the interpretation of results in Figure 10, let us assume that we have two disjoint subgraphs with a large number of potential missing edges between them. Then, handling such the missing edge problem means that, despite their lack of explicit connections, two subgraphs share meaningful knowledge with each other, which leads to the joint improvement of models that train on them (i.e., the performance of (Local) and (Neighbor) models are both high). In other words, failure in handling the missing edge problem means that there is little/no information sharing between interconnected subgraphs with a large number of missing edges, and consequently, the performances of both (Local) and (Neighbor) models are low. From this statement, since our FED-PUB outperforms all the other methods on both (Local) and (Neighbor) measures in Figure 10, we can sufficiently say that our FED-PUB facilitates the information sharing between relevant subgraphs, which then helps alleviate the missing edge problem.
>
> ---
>
> **Question 5:** What is the meaning of J in Eq. (2)? If J is the number of subgraphs within a community, how to find a specific community for each subgraph during model training? And does that mean the aggregation is only executed within the subgraphs in a community?
>
> **Answer:** We apologize for the confusion. J is the total number of subgraphs (i.e., the total number of FL participants). Also, as described in Lines 218-221, we perform weight averaging of each model based on its functional similarity to all the other models, as totally ignoring the model weights from other communities may result in exploiting only the local objective while ignoring the globally useful weights.
>
> ---
>
> **Question 6:** In Line 45, the authors claim that the algorithms in [40] and [43] may compromise data privacy. Is there any further evidence to prove that?
>
> **Answer:** Yes, there is evidence to prove that [40] and [43] may compromise data privacy. Let us assume that sending the local client data/representation to the server or other clients is evidence of data privacy violation in federated learning, from the observation of the work [A]. Then, from this assumption, [40] and [43] clearly violate the data privacy constraints during local graph expansion processes (Lines 41-44), for the following reasons:
>
> * In [40], to augment the similar nodes stored in the other clients as the neighboring nodes in the current client’s subgraph, they share the nodes (i.e., node features) between clients. In particular, the client first sends its nodes to the server, and the server distributes the similar nodes located in the other clients to that client who sent their nodes to the server, which violates the privacy constraints. Also, in Section 3.5 of the authors’ paper [40], the authors already described the potential data privacy issues of their methods.
>
> * In [43], to generate the features of similar nodes stored in the other clients and then augment them in the current client’s subgraph, they share the node representations between subgraphs. More specifically, to generate the realistic node features used for subgraph expansion, this work [43] trains to maximize the similarities between the generated node representations in the certain client and the initial node representations in the other clients with their representation sharing, in which data privacy violation may occur.
>
> [A] Sun et al. Soteria: Provable Defense against Privacy Leakage in Federated Learning from Representation Perspective. CVPR 2021.

---

> ### Author Response · Authors · 2022-07-31
> **Initial Response (3/5) to Reviewer gnhR**
>
> **Question 3.1:** Even though the authors use an intuitive explanation and a tiny experiment on the synthetic dataset in Figure 1 to demonstrate the heterogeneity problem and further show the need for personalized subgraph FL, the motivation behind personalized subgraph FL is not convictive enough on real-world datasets. How serious the heterogeneity problem is on real-world datasets under the subgraph FL setting?
>
> **Answer:** The heterogeneity is highly problematic in real-world datasets of personalized subgraph FL, especially when there are lots of FL participants with different knowledge. Please note that we already experimentally demonstrated this issue in Table 1 and Table 2 that, due to the heterogeneity between subgraphs, global weight aggregation methods (e.g., FedAvg) perform poorly when the number of clients is large since they collapse the incompatible knowledge into one global model, while our FED-PUB is not.
>
> However, as suggested, we have further quantified the exact amount of heterogeneities by varying the number of clients. Specifically, for each subgraph, we first measure its distance to other subgraphs by measuring their Jenson-Shannon divergence of label distributions, and then report the median divergence value between subgraphs. Thus, if the median divergence values are large, then we can regard that the datasets are heterogeneous enough to cause incompatible knowledge issues. We report such divergence values on the Cora, CiteSeer, and PubMed datasets of the nonoverlapping node scenario with 3, 5, 10, and 20 clients, in the table below. Then, based on the reported results in the table, we can confirm that, when we increase the number of clients, the heterogeneity across clients becomes more severe and problematic for personalized subgraph FL, and thus is an important issue to tackle.
>
> | # Clients | Cora | CiteSeer | PubMed |
> | --- | --- | --- | --- |
> | 3 | 0.520 | 0.437 | 0.373 |
> | 5 | 0.590 | 0.517 | 0.362 |
> | 10 | 0.606 | 0.541 | 0.392 |
> | 20 | 0.665 | 0.568 | 0.424 |
>
> ---
>
> **Question 3.2:** Regarding heterogeneity, what if the number of clients is much smaller than 10, which may show the significance of personalized subgraph FL across different heterogeneities?
>
> **Answer:** Following your suggestion, to see the performance of models when the heterogeneity issue is less significant, we have conducted experiments in the setting where the number of clients is smaller than 10 (i.e., 3), on the Cora, CiteSeer, and PubMed datasets of the nonoverlapping node scenario. As shown in the table below, compared to the results in Table 2 with numbers of clients as 5, 10, and 20, the performance gaps between our FED-PUB and baselines are much reduced. However, we clearly observe that our FED-PUB meaningfully (i.e., statistically) outperforms all the other baselines even when the number of clients is small, since there still exists incompatible knowledge across clients, which our FED-PUB effectively tackles with personalized weight aggregation and local weight masking schemes.
>
> | Model | Cora | CiteSeer | PubMed |
> | --- | --- | --- | --- |
> | Local | 81.73 ± 0.44 | 68.16 ± 0.25 | 84.81 ± 0.40 |
> | FedAvg | 78.77 ± 0.13 | 69.34 ± 0.23 | 85.29 ± 0.20 |
> | FedProx | 78.91 ± 0.21 | 69.54 ± 0.27 | 85.59 ± 0.18 |
> | FedPer | 82.29 ± 0.13 | 69.80 ± 0.33 | 85.34 ± 0.16 |
> | FedGNN | 82.36 ± 0.62 | 67.79 ± 0.49 | 85.57 ± 0.13 |
> | FedSage | 77.79 ± 1.96 | 69.35 ± 0.12 | 85.63 ± 0.22 |
> | GCFL | 82.67 ± 0.74 | 68.85 ± 0.58 | 86.20 ± 0.15 |
> | FED-PUB (Ours) | **84.45** ± 0.23 | **70.66** ± 0.34 | **86.74** ± 0.16 |

---

> ### Author Response · Authors · 2022-07-31
> **Initial Response (2/5) to Reviewer gnhR**
>
> **Question 2.1:** The relationship between the weight masking method and the research problem is unclear. The authors claim that the weight masking mechanism is to select the relevant parameters for each subgraph. However, it seems like a general solution for personalized FL under heterogeneity and is less relevant to the target subgraph FL problem.
>
> **Answer:** As you mentioned, our weight masking scheme may be also helpful for general FL. However, it is more helpful and relevant to our subgraph FL problems since **heterogeneity arising in personalized subgraph FL is more severe** than in general FL scenarios, that is subgraphs in graph FL may be **completely disjoint**, while in general FL each client is likely to train on shifted, but similar data distributions. Also, please see the results in Figure 7 (ablation study), which shows that the weight masking scheme largely improves the performance on the subgraph FL tasks when all the subgraphs are disjoint (i.e., there exist extreme distribution shifts between subgraphs), with a much larger performance gain over the gain obtained on the overlapping node scenario, which shows that the weight masking is a necessary design choice for subgraph FL.
>
> ---
>
> **Question 2.2:** Although weight masking brings some performance improvement, how it benefits the node classification problem is unknown.
>
> **Answer:** To further quantify how the weight masking scheme benefits the node classification task, we have additionally measured how much distributional shifts exist among the subgraphs. In particular, we have measured the label differences between subgraphs with the Jenson-Shannon divergence, in which the minimum and maximum values are 0 and 2, respectively, on the Cora dataset with 20 different clients over the overlapping and the non-overlapping scenarios. Then, we observe that the distance (i.e., divergence value) among the subgraphs within the same community is 0.384 while the distance among the subgraphs belonging to different communities is 0.639 for the non-overlapping node scenario. On the other hand, the distance among the subgraphs within the same community is 0.047 while the distance among the subgraphs belonging to different communities is 0.528 for the overlapping node scenario. Thus, from the fact that **heterogeneity of subgraphs within the same community is extremely larger in the non-overlapping setting (0.384)** compared to the overlapping setting (0.047), personalized weight aggregation might not be enough in disjoint subgraph FL problems, and, as shown in Figure 7, **weight masking scheme is clearly required** to selectively utilize and train only on the relevant parameters to each local task.

---

> ### Author Response · Authors · 2022-08-02
> **Initial Response (1/5) to Reviewer gnhR**
>
> We sincerely thank you for your constructive and helpful comments. We appreciate your positive comments that our research problem is interesting and novel yet challenging, and that the experiments are extensive which proves the effectiveness of our method. We initially address all your concerns below:
>
> ---
>
> **Question 1:** Despite some interesting designs in the proposed model, the core idea is not novel enough and similar to some existing works, which either use shared data to minimize the distance between neural networks of different sizes [*1] or incorporate auxiliary data to assist the federated training procedure with knowledge distillation [*2, *3, *4].
>
> **Answer:** Thank you for pointing out relevant works. However, our main novelty comes from proposing the **novel personalized subgraph FL task** with an undiscovered challenge of **community structures**, along with the **missing edge problem**, for **subgraph FL domains**, and we tackle those unique challenges in subgraph FL with our personalized weight aggregation and local weight masking schemes. From this perspective, we strongly believe that our work is sufficiently novel, although some of its components may be reminiscent of the ones for other FL settings. Moreover, even from the methodological perspective, our work significantly differs from the suggested works, which we describe one by one as follows:
>
> 1. [*1] This recent work aims to aggregate neural networks of different shapes, and, for which, the authors propose to minimize the distances between the model representations obtained by forwarding the same input, since aggregating the neural network weights of different sizes is not straightforward in FL. However, compared to this work, we aim to tackle our novel personalized subgraph FL problem where there particularly exist community structures, and, for which, we propose to perform personalized weight aggregation between clients based on their models’ functional similarities obtained from the same randomly generated graph. Therefore, our work not only targets a completely different problem -- aggregating different-sized neural models vs capturing community structures for personalized subgraph FL -- but also uses the obtained representations differently: minimizing the differences between the obtained representations as a training objective vs identifying the similar subgraphs to largely share weights among them. Furthermore, please note that we already acknowledge existing works on functional embeddings in Lines 203-210. However, our work differs in how we utilize functional embeddings for tackling the unique challenges in subgraph FL tasks.
>
> 2. [*2, *3, *4] Those works utilize the knowledge distillation in FL, in which the local model outputs are used as auxiliary data to train the other server/client models. However, they are largely different from ours: we obtain the models’ functional embeddings to largely share weights between similar subgraphs, and our work does not aim to generate/predict auxiliary labels for additional training of the neural models in the knowledge distillation process.
>
> Thus, to summarize our main novelty, we study the novel problem of personalized subgraph FL where different subgraphs with similar properties are likely to belong to the same community while edges are missing between them, and we tackle this challenging problem with both the personalized weight aggregation scheme based on models’ functional embeddings and the local weight masking scheme with its sparsification. We believe that both are sufficiently novel and effective solutions for the novel problem of personalized subgraph FL.

---

> ### Author Response · Authors · 2022-08-07
> **The end of the discussion phase is approaching.**
>
> Dear Reviewer gnhR,
>
> We sincerely appreciate your positive comments: our research problem is novel and challenging, which can inspire future studies, and the experiments conducted are extensive, which proves the effectiveness of our methods.
>
> However, you pointed out the novelty of our work by providing relevant works, but also suggested to us clarifying the advantage of weight masks, the heterogeneity issue on real-world datasets, and the claim on missing edge problems. To this end, during the rebuttal period, we have made every effort to faithfully address all of them, which are given in detail in the responses below. In short, we have concretely discussed which of ours are different from the suggested works, i.e., we formalize a new problem in subgraph FL based on two challenges of missing edges and network communities and then propose to identify the interconnected subgraphs with functional embeddings, which are our main novelty. Furthermore, we have provided sufficient evidence with experimental results that weight masking is significantly helpful to our subgraph FL problem, heterogeneity issues are severe in real-world datasets, and missing edge problems are indeed implicitly handled by our FED-PUB.
>
> In the end, we are confident that we have fully addressed all of your comments/concerns in the initial responses. Therefore, since the discussion phase will close soon, could you please go over our complete responses? Please let us know if you have anything else that we should address.
>
> Best regards, Authors

---

> > ### Comment · Reviewer_gnhR · 2022-08-09
> > **Reply to the response of review**
> >
> > I've read all the responses. Some of my questions are addressed by these explanations and additional experiments.
> >
> > However, even if the authors have pointed out the minor difference between the proposed work and existing methods, I still hold my view that the novelty of the proposed method is limited. Although there are some differences, the core mechanism of the proposed method is still similar to some classic FL methods. Another concern is about the weight masking mechanism, which I still think this technique lacks a strong motivation behind and seems inconsistent with the learning task.
> >
> > Considering the authors' responses have addressed some of my concerns, I would like to change my rating from 4 to 5. I think the paper can be better if more novel and well-motivated techniques can be applied here.

---

> > > ### Author Response · Authors · 2022-08-09
> > > **Thank you for your reply**
> > >
> > > We sincerely appreciate all your constructive and helpful comments, and also your reply to our initial responses. We are happy to hear that, throughout our initial responses with additional experiments, some of your questions are addressed.
> > >
> > > ---
> > >
> > > However, you are still concerned about the minor differences between our proposed FED-PUB and existing works, and here we would like to faithfully address this concern once again. We believe the big difference in our work stems from our motivations. Specifically, we handle two critical challenges in subgraph FL: missing edge and network community problems, by formalizing a new personalized subgraph FL problem, which is clearly different from existing works. Our architecture is also oriented to tackle this personalized subgraph FL problem, which is divided into two sub-problems: overlapping and nonoverlapping subgraphs. In particular, to promote knowledge sharing between the overlapping subgraphs, we propose to share weights largely between the interconnected subgraphs (Figure 6 and Figure 7). On the other hand, to deal with the heterogeneity issues when subgraphs are completely disjoint, we propose to train personalized weight masks that are not shared across different clients (Figure 7 and Figure 8). Thus, in terms of the main novelty, our work should be evaluated more on its contributions to discovering challenges and solving them in subgraph FL domains, and we would like to politely claim that, considering all these perspectives above, ours are valuable and well-motivated.
> > >
> > > ---
> > >
> > > Also, the one remaining concern you have is that the proposed weight masking mechanism lacks strong motivation; however, we believe the weight masking scheme is the necessary design choice for the subgraph FL problem. In particular, as described in the previous responses (responses to Question 2.1 and Question 2.2), the heterogeneity issue of subgraphs is severe when subgraphs are completely disjoint. And this issue is not easily solvable via personalized weight aggregation, since there are no good subgraphs that provide helpful weights to the highly heterogeneous subgraph. Then, to tackle this issue, personalized weight masking should be used, since this scheme can filter out irrelevant information transmitted from the other heterogeneous subgraphs, while allowing the model to maintain the locally helpful information in its parameters. Note that the severity of heterogeneity for nonoverlapping subgraphs was additionally provided in Question 2.2, and we already provided the advantage of using weight masking schemes in Figure 7, Ablation studies.

---

### Official Review · Reviewer_cN7m · 2022-07-10

**Rating:** 5
**Confidence:** 3
**Soundness:** 3 good
**Presentation:** 3 good
**Contribution:** 3 good

**Summary:**

This paper proposes an algorithm for personalized FL subgraph learning, FED-PUB.

The server compute similarities among clients based the output of the local GNNs using random graphs as inputs. It then use them to perform weighted averaging for models.

Each client also learns a personalized sparse mask to select and update only part of the aggregated parameters.

Experiments show the performance.

**Questions:**

1. Why not directly use local node label distributions to calculate the similarities among clients? The authors could add a comparison with this method.

2. The non-overlapping node scenario is more challenging since it is more heterogeneous. Why does FedAvg (Table 2) in non-overlapping have better performance than FedAvg (Table 1) in overlapping.

3. In Appendix B.3, for small datasets, namely Cora, CiteSeer and PubMed, the experiments set the number of local training epoch as 1. The setting is equal to distributed training. Why does FedAvg perform poorly in this setting? Is it due to missing edges?





**Limitations:**

The intuition of weighted aggregation by similarity has been widely adopted. But the mechanism of similarity calculation by a single random input is not convincing to me. A performance comparison with other similarity methods (label similarity, gradient similarity, model similarity) will also be very helpful.

**Strengths And Weaknesses:**

Strengths:
1. The weighted aggregation based on similarities of outputs of random graphs is interesting.
2. The personalized sparse mask looks helpful.
3. Code is provided.

Weaknesses:
1. The advantages by using outputs of random graphs are not convincing to me. The random graph based on SBM has too much randomness. Especially with random node features with normal distribution.
2. The code only provides fedavg and fedpub.

---

> ### Author Response · Authors · 2022-07-31
> **Initial Response (3/3) to Reviewer cN7m**
>
> **Question 5:** In Appendix B.3, for small datasets, namely Cora, CiteSeer, and PubMed, the experiments set the number of local training epochs as 1, which is equal to distributed training. Why does FedAvg perform poorly in this setting? Is it due to missing edges?
>
> **Answer:** Yes, it is due to the problem of missing edges, which further negatively affects the incompatible knowledge issue. To see how much the missing edges yield the performance drops in subgraph FL, we first train a FedAvg on the connected global graph and then evaluate it on disjoint subgraphs over all the clients, on the Cora dataset of both Non-overlapping and Overlapping node scenarios with varying client numbers. Note that this setting (i.e., Oracle) is unfair in comparison to all the other FL methods, since this Oracle model can observe the missing edges during training, while all the other methods are not.
>
> | Model | NonOverlapping-5 | NonOverlapping-20 | Overlapping-10 | Overlapping-50 |
> | --- | --- | --- | --- | --- |
> | Oracle | 85.07 | 85.47 | 85.08 | 85.28 |
> | Local | 81.30 | 80.30 | 73.98 | 76.63 |
> | FedAvg | 74.45 | 69.50 | 76.48 | 53.99 |
> | FedGNN | 81.51 | 70.10 | 70.63 | 56.91 |
> | FedSage | 72.97 | 57.97 | 77.52 | 55.48 |
> | FED-PUB (Ours) | 83.70 | 81.75 | 79.60 | 77.84 |
>
> As shown in the table above, we can observe that the Oracle model outperforms all the other methods, which brings us to the following points. At first, due to the problem of missing edges, all the FL methods, which observe edges only within each subgraph, perform poorly than the Oracle method. Also, the missing edge problem negatively affects the incompatible knowledge issue: since all client models are trained by the partial subgraphs, which are parts of the larger global graph, the trained parameters in the client and the aggregated parameters in the server might not capture globally meaningful knowledge or the knowledge that is helpful to the other clients, which explains why the FedAvg performs poorly in this subgraph FL setting.

---

> ### Author Response · Authors · 2022-07-31
> **Initial Response (2/3) to Reviewer cN7m**
>
> **Question 3.1:** Why not directly use local node label distributions to calculate the similarities among clients?
>
> **Answer:** The usage of label distributions stored in the private local subgraphs may violate the privacy constraint in FL. Thus, in this work, we do not use such local information when calculating the similarities between clients. However, as shown in Figure 6, the functional similarities of subgraphs calculated by our FED-PUB are close to the similarities calculated by the label distributions, which verifies that ours can capture similarities of label distributions among subgraphs without using their local labels.
>
> ---
>
> **Question 3.2:** A performance comparison with other similarity methods (label similarity, gradient similarity, model similarity) will also be very helpful.
>
> **Answer:** Please note that, in Figure 3, we already compared the effectiveness and efficiency of parameter, gradient, and functional similarities, and then showed that parameter and gradient similarities are neither effective nor efficient compared to our functional similarity. Also, as we answer in the previous question (Question 3.1) above, we already showed that the similarities of subgraphs from our functional embeddings are similar to their label similarities in Figure 6.
>
> However, as suggested, we have additionally conducted experiments on the parameter, gradient, and label similarities, on the Cora dataset of the overlapping node scenario with the number of clients as 30, and then reported the results on 20, 40, 60, and 80 epochs in the table below.
>
> | Model | 20 | 40 | 60 | 80 |
> | --- | --- | --- | --- | --- |
> | FedAvg | 29.94 | 32.69 | 47.84 | 52.42 |
> | Parameter | 29.94 | 35.89 | 47.03 | 52.28 |
> | Gradient | 33.93 | 51.09 | 52.77 | 58.14 |
> | Label | 65.97 | 74.31 | 76.50 | 76.82 |
> | Function (FED-PUB) | 67.82 | 73.51 | 74.66 | 75.90 |
>
> As shown in the above table, the models, which utilize the parameter and gradient for calculating the similarities between subgraphs, perform similarly to the FedAvg model and perform worse than our functional and label similarity schemes. However, even though the label similarity model uses privacy-sensitive local information (i.e., label distributions of every client), the performance of our FED-PUB that utilizes the functional embeddings is similar to the performance of the label model. Therefore, along with the results in Figure 6, this comparison results between similarity schemes further verify the effectiveness of our functional embedding scheme in capturing the similarities among subgraphs.
>
> ---
>
> **Question 4:** The non-overlapping node scenario is more challenging since it is more heterogeneous. Why does FedAvg (Table 2) in non-overlapping have better performance than FedAvg (Table 1) in overlapping?
>
> **Answer:** The performances in Table 1 for the overlapping node scenario and Table 2 for the non-overlapping node scenario are not comparable, since the subgraphs used for training and evaluation are different due to differences in experimental settings, which are described in Lines 266-270 of the main paper, Section B of the supplementary file, and Table 1 of the supplementary file. However, in general, when the number of clients increases, the knowledge collapse issue of FedAvg becomes more problematic. Therefore, since overlapping node scenarios have more numbers of clients than the non-overlapping node scenarios in Table 1 and Table 2 (i.e., overlapping node scenarios have client numbers of 10, 30, and 50, meanwhile, non-overlapping node scenarios have client numbers of 5, 10, and 20), FedAvg in overlapping node settings underperforms than in non-overlapping node settings.

---

> ### Author Response · Authors · 2022-08-02
> **Initial Response (1/3) to Reviewer cN7m**
>
> We sincerely thank you for your constructive and helpful comments. We appreciate your positive comments that personalized weight aggregation based on functional embedding is interesting, personalized sparse masks look helpful, and we provide the code for our FED-PUB. We initially address all your concerns below:
>
> ---
>
> **Question 1.1:** The advantages of using the outputs of random graphs are not convincing to me.
>
> **Answer:** There are obvious advantages of using the outputs of random graphs (i.e., functional embeddings), summarized as follows:
> * As described in Lines 198-202 with Figure 3, compared to the parameter and gradient similarities, which are measured on high-dimensional spaces but those spaces are neither efficient nor meaningful due to the curse of dimensionality [5], our functional similarities work on low-dimensional spaces with both high efficacy and efficiency.
> * As shown in Figure 6, our functional embeddings obtained by outputs of random graphs accurately capture the similarity (community) structures between subgraphs.
>
> ---
>
> **Question 1.2:** The random graph based on SBM has too much randomness. Especially with random node features with normal distribution.
>
> **Answer:** The random graph based on SBM is initially at once (i.e., not randomly changed at every iteration), and randomly initializing node features with normal distribution is more beneficial than using the pre-defined node features.
>
> Specifically, as described in B.3 Implementation Details of the supplementary file, to calculate each client model’s functional embedding, we use the same random graph for all clients, by first initializing the random graph at once in the server and then distributing it to all clients. Also, using the randomly initialized features is more helpful when capturing the functional space, since randomness does not yield any bias on that space, while particularly initialized node features from exiting nodes and their label distributions might have a bias toward particular subspaces. To experimentally validate this statement, we compare various schemes used for calculating the functional embeddings: 1) SBM denotes the random graph generated from the SBM model like ours; 2) ER denotes the random graph generated from the Erdos-Renyi model; 3) One denotes the random graph having only one node; 4) Feature denotes the graph where nodes are initialized by the node features in the client. We then measure the performances of those four schemes by calculating the correlation coefficients between label distributions and generated similarities of subgraphs (i.e., the high value means that the similarities from the functional embeddings are similar to the label distributions) on the Cora dataset of non-overlapping and overlapping node scenarios with 20 clients, which are reported in the table below.
>
> | Graph | Overlapping | Non-Overlapping |
> | --- | --- | --- |
> | SBM | 0.937 | 0.810 |
> | ER | 0.920 | 0.712 |
> | One | 0.822 | 0.656 |
> | Feature | 0.897 | 0.632 |
>
> As shown in the table, compared to the One scheme that uses only one node for calculating the functional embeddings, SBM and ER schemes that use more large numbers of randomly initialized nodes can accurately capture the similarities between subgraphs. This result confirms that a sufficient amount of randomness is required to identify the model’s functional space. Also, compared to the Feature scheme that uses existing node representations for calculating the functional embeddings, SBM and ER random models show superiority in capturing similarities among subgraphs, which verifies that randomness might help obtain accurate functional embeddings of the models without bias.
>
> ---
>
> **Question 2:** The code only provides FedAvg and FED-PUB.
>
> **Answer:** We would like to politely speak that this point cannot be our weakness, since we not only provide our FED-PUB code in the supplementary material but also we faithfully follow the available codes of existing models provided in their papers. To mention a few, for subgraph FL works, we use the following authors’ codes:
> * FedGNN: https://github.com/wuch15/FedPerGNN
> * FedSage: https://github.com/zkhku/fedsage

---

> ### Author Response · Authors · 2022-08-07
> **The end of the discussion phase is approaching.**
>
> Dear Reviewer cN7m,
>
> We sincerely appreciate your positive comments that the personalized weight aggregation based on functional embeddings is interesting, the sparse weight masking looks helpful, and we provide our source code.
>
> On the other hand, you pointed out two weaknesses: the advantage of using random graphs for functional embeddings is unclear; we only provide the source code for our FED-PUB and baseline FedAvg models. In the responses below, we have clearly validated with additional experiments that, instead of using node features stored in clients, randomly initializing graphs as the inputs for functional embeddings is more helpful since they might not yield any bias on the model’s functional space. Also, regarding random graphs, there is an additional advantage that we can preserve the privacy of FL participants’ local data. Lastly, we would like to politely say again that providing the codes for FedAvg and FED-PUB is our strength, which should not be the weakness of our work.
>
> In the end, we are confident that we have fully addressed all of your comments/concerns in the initial responses. Therefore, since the discussion phase will close soon, could you please go over our complete responses? Please let us know if you have anything else that we should address.
>
> Best regards, Authors

---

> > ### Comment · Reviewer_cN7m · 2022-08-09
> > **Reply to the response of review**
> >
> > I want to thank the authors for the detailed response. My concerns on random graphs and similarity computation have been resolved.
> >
> > However, as shown in the answer of **Question 3.2: A performance comparison with other similarity methods (label similarity, gradient similarity, model similarity) will also be very helpful.** The label similarity has the same or better performance than the proposed method, since the proposed method is trying to make it close to the label similarity. The label distribution might not be privacy sensitive (It is not directly related with user information and traditional FL algorithms use it for similarity calculation) and also can be resolved by privacy preserve methods.
> >
> > The idea on using random graphs for functional embedding is interesting. I will keep my score and I also think the paper can be better if more novel and well-motivated techniques can be applied here.

---

> > > ### Author Response · Authors · 2022-08-09
> > > **Thank you for your reply**
> > >
> > > We sincerely appreciate all your comments and questions, as well as your reply to our initial responses. Also, thank you for acknowledging our detailed responses. We are happy to hear that your concerns about random graphs and similarity computations have been resolved.
> > >
> > > ---
> > >
> > > However, we would like to clarify your remaining concern about the label similarity. Note that the goal of federated learning is to leave the local data on the client side while learning the model by aggregating locally-computed updates, to preserve the privacy of the user's data. Then, in terms of privacy-preserving, **it is more practical and safer not to share any type of local data, including features, labels, and their distributions**, compared to sharing some of them. This privacy issue on sharing label distributions becomes more severe in heterogeneous settings with Non-IID label distributions (i.e., non-overlapping subgraphs). For example, let's assume all nodes in client A have the label 0, while all nodes in the other client B have the label 1. Then, the server can know which clients have which node labels, which are not optimal in federated learning, especially when we are concerned about user privacy. Therefore, in our work, we consider more rigorous federated learning scenarios, where any types of local data related to the client's data instances are not sharable and should only be stored on the client side.
> > >
> > > As you pointed out, one might use privacy-preserving methods, such as differential privacy techniques, and they might make federated learning systems safer even when we share local information (e.g., local data distributions). However, investigating how we can preserve the privacy of label distributions with differential privacy methods is beyond the scope of our work, and we leave it as future work.
> > >
> > > ---
> > >
> > > Last but not least, regarding novel and well-motivated techniques, we are confident that we motivate our personalized subgraph FL problems with two important challenges: missing edge and network community problems, and we tackle our novel task with personalized weight aggregation, which is particularly helpful for overlapping scenarios, and with personalized weight mask, which is particularly helpful for non-overlapping scenarios. Thus, we would like to politely claim that our current contributions are well-motivated and valuable.

---

### Official Review · Reviewer_we3U · 2022-07-11

**Rating:** 4
**Confidence:** 5
**Soundness:** 2 fair
**Presentation:** 3 good
**Contribution:** 2 fair

**Summary:**

This paper introduces a personalized FL framework to train graph neural network model across subgraphs owned by different clients. The authors argue that the heterogeneity of subgraphs will cause the unexpected knowledge collapse when using the same global model to fit all the data from different subgraphs. The authors first group subgraphs into different communities and then train personalized model parameters for clients in each community by introducing adaptive aggregation weights and unique parameter masks. Experimental results support the advantage of the proposed method FED-PUB on six datasets.

**Questions:**

See Weakness.
Plus, why not compare with FedSage+ [43]?

**Limitations:**

See Weakness.

**Strengths And Weaknesses:**

Strengths
	Compared with existing popular methods, the proposed framework achieves competitive results on six popular federated graph benchmarks in a more communication efficient and privacy-preserved manner, without sharing features or embeddings of nodes in graphs.
	The method to identify the similarity between subgraphs is novel, which uses input-based functional similarity instead of the inner of model parameters (i.e. cossim) to estimate the communities of user data.
	The proposed method can potentially handle the missing edges problem by only consider the model aggregation in the same community for each client, since the edges and neighbors are dense in the same community.

Weaknesses
	This work tackles the heterogeneity of different subgraphs by mitigating incompatible knowledge collapse when using only one global model to fit all the data. However, the explaination for this incompatibility between different communities is not adequate. I wonder whether the too small model size or the missing edges between nodes cause such incompatibility. Maybe the direct model training on the whole datasets is needed as the oracle case to show such incompatibility.
	The proposed method is not validated in the cases where the nodes and edges of the whole graph are uniformly distributed across clients. Due to the complex distribution of graph data in practice, the performance of the method in such case or cases where there are not obvious communities should be also considered.

---

> ### Author Response · Authors · 2022-07-31
> **Initial Response (2/2) to Reviewer we3U**
>
> **Question 1.3:** I wonder whether the missing edges between nodes or the too small model size cause incompatibility between subgraphs.
>
> **Answer:** Yes, the problem of missing edges is one of the factors that cause such the knowledge incompatibility issue as explained before (See the previous answer for Question 1.2). Also, as you mentioned, increasing the model sizes may alleviate the knowledge incompatible issue to some degree. However, this is largely suboptimal since such a large model may suffer from overfitting and large computation and communication overheads. Also, our FED-PUB with personalized modules still outperforms the global aggregation model, regardless of the model sizes.
>
> Specifically, we have varied the dimensionality of the hidden layers for FedAvg and our FED-PUB in the range of {64, 128, 256, 512, 1024, 2048} on the Cora dataset of the overlapping node scenario with the number of clients as 30, and then reported the results in the table below. The results show that when we increase the number of dimensions from 64 to 256, the performances of FedAvg which aggregates all local models into the global model improve. However, when we further increase the hidden dimension sizes, the FedAvg does not yield any further performance improvements, and when using the largest hidden dimensionality, it suffers from a performance drop due to the overfitting issues. Note that our FED-PUB significantly outperforms FedAvg regardless of the hidden dimensionality, which verifies that simply increasing the model capacity does not effectively solve the knowledge collapse issues. On the other hand, our personalized subgraph FL framework with personalized weight aggregation and parameter masking effectively handles such challenges.
>
> | Model | 64 | 128 | 256 | 512 | 1024 | 2048 |
> | --- | --- | --- | --- | --- | --- | --- |
> | FedAvg | 50.75 | 53.99 | 61.13 | 60.99 | 61.26 | 59.12 |
> | FED-PUB (Ours) | 74.64 | 75.40 | 74.86 | 74.96 | 74.60 | 73.84 |
>
> ---
>
> **Question 2:** The proposed method is not validated in cases where the nodes and edges of the whole graph are uniformly distributed across clients. Due to the complex distribution of graph data in practice, the performance of the method in such cases where there are no obvious communities should be also considered.
>
> **Answer:** Please note that uniform partitions of graphs are more unrealistic, and that is why we used graph partitioning algorithms (e.g., the METIS algorithm). For example, when we partition the entire graphs of the Cora, CiteSeer, and PubMed datasets into 10 different subgraphs uniformly at random, the number of nodes per subgraph will be larger than the number of edges (e.g., some nodes do not have any edges) as shown in the below table, which is uncommon in practice.
>
> | Dataset | # Nodes | # Edges |
> | --- | --- | --- |
> | Cora | 248 | 97.4 |
> | CiteSeer | 212 | 72.0 |
> | PubMed | 1971 | 884.6 |
>
> However, as suggested, we have additionally conducted experiments on the random split setting with 10 different clients on the CiteSeer dataset, and then provided the results in the below table. As shown in the table, while the gap between baselines and our model is reduced compared to the non-overlapping and overlapping scenarios since there is no specific community structure in this random setting, our FED-PUB significantly outperforms all the other baselines under this setting as well.
>
> | Model | Random with # clients 10 |
> | --- | --- |
> | Local | 44.27 ± 1.05 |
> | FedAvg | 60.84 ± 0.80 |
> | FedProx | 59.38 ± 1.66 |
> | FedPer | 60.04 ± 0.93 |
> | FedGNN | 54.64 ± 1.67 |
> | FedSage | 61.03 ± 0.11 |
> | GCFL | 53.15 ± 1.82 |
> | FED-PUB (Ours) | **63.63** ± 0.86 |
>
> ---
>
> **Question 3:** Why not compare with FedSage+ [43]?
>
> **Answer:** We apologize for the confusion. As described in B.2 Baselines and Our Model Section of the supplementary file, we compare FedSage+ [43], and we refer to this FedSage+ model as FedSage in our work. We will correct this in the next revision.

---

> > ### Comment · Reviewer_we3U · 2022-08-10
> > **Reply**
> >
> > I want to thank the authors for the detailed response. My concerns  have almost been resolved.
> >
> > The idea is interesting. I will raise my score to borderline accept. I am also aware that the paper can be better if more novel and well-motivated techniques can be applied here.

---

> > > ### Author Response · Authors · 2022-08-10
> > > **Thank you for your reply**
> > >
> > > We sincerely thank you for your helpful comments, as well as your time and effort in reviewing our paper. We are happy to hear that your concerns are mostly resolved.
> > >
> > > **NOTE:** I know you must be very busy, but could you please reflect your updated score in your original comment? You promised to increase your score, but it seems the updated score is not reflected in the system yet.
> > >
> > > ---
> > >
> > > Regarding your last comment that our paper can be better if more novel and well-motivated techniques can be applied here, throughout constructive discussions with reviewers, we are confident that our work is sufficiently novel and well-motivated. In particular, we motivate our novel personalized subgraph FL problem with two important challenges: missing edge and network community problems, and we tackle our novel task with personalized weight aggregation, which is particularly helpful for overlapping scenarios, and with personalized weight mask, which is particularly helpful for non-overlapping scenarios. Thus, the task that we formalize is important for subgraph FL, and the ingredients that we propose are also necessary for our formalized task. In this vein, we would like to politely argue that our current contributions are well-motivated and valuable.
> > >
> > > ---
> > >
> > > Once again, thank you for reading our responses and replying to us.

---

> ### Author Response · Authors · 2022-08-02
> **Initial Response (1/2) to Reviewer we3U**
>
> We sincerely thank you for your constructive and helpful comments. We appreciate your positive comments that our functional embedding method to identify similar subgraphs is novel, our FED-PUB can tackle the missing edge problem, and ours outperforms existing subgraph benchmarks in a more communication-efficient and privacy-preserved way. We initially address all your concerns below:
>
> ---
>
> **Question 1.1:** This work tackles the heterogeneity of different subgraphs by mitigating the incompatible knowledge collapse issue, which is occurred when using only one global model to fit all the data. However, the explanation for this incompatibility is not adequate.
>
> **Answer:** The incompatibility issue is sufficiently explained in our motivation figures and main results, and here we explain again why the incompatibility issue arises in subgraph FL. At first, as illustrated in Figure 1 (B) and Figure 1 right, an incompatible knowledge collapse clearly happens for the model that aggregates the knowledge from different communities at different clients into one. Moreover, the main results in Table 1 and Table 2 support this claim. Specifically, in those two tables, FedAvg often underperforms the local model without parameter aggregation, especially when we increase the number of clients, since FedAvg collapses incompatible knowledge from different subgraphs (communities) into one while the local models preserve the local knowledge in their parameters. However, since our FED-PUB can further share compatible and meaningful knowledge among subgraphs in the same community with personalized weight aggregation while filtering out irrelevant knowledge for the local task with local weight masking, ours significantly outperforms those baselines in such a challenging setting of the incompatibility between different subgraphs.
>
> ---
>
> **Question 1.2:** Maybe direct model training on the whole dataset is needed as the oracle case to show such incompatibility.
>
> **Answer:** Thank you for your suggestion. Please note that direct training on the whole dataset results in unfair comparison due to missing edges: this oracle setting can observe missing edges between subgraphs during training, while all the other methods cannot observe them.
>
> However, as suggested, we have conducted an additional experiment that we train a model on the connected global graph and then evaluate it on disjoint subgraphs over all clients, on the Cora dataset of both Non-overlapping and Overlapping node scenarios with varying client numbers. As shown in the below table, the Oracle model outperforms all the other methods, while our FED-PUB achieves the closest performance to the Oracle model.
>
> | Model | NonOverlapping-5 | NonOverlapping-20 | Overlapping-10 | Overlapping-50 |
> | --- | --- | --- | --- | --- |
> | Oracle | 85.07 | 85.47 | 85.08 | 85.28 |
> | Local | 81.30 | 80.30 | 73.98 | 76.63 |
> | FedAvg | 74.45 | 69.50 | 76.48 | 53.99 |
> | FedGNN | 81.51 | 70.10 | 70.63 | 56.91 |
> | FedSage | 72.97 | 57.97 | 77.52 | 55.48 |
> | FED-PUB (Ours) | 83.70 | 81.75 | 79.60 | 77.84 |
>
> The above results bring us to the following points. In particular, due to the problem of missing edges, all the FL methods, which observe edges only within each subgraph, perform poorly than the Oracle method. Also, the missing edge problem negatively affects the incompatible knowledge issue: since all client models are trained by the partial subgraphs, which are parts of the larger global graph, the trained parameters in the client and the aggregated parameters in the server might not capture globally meaningful knowledge or the knowledge that is helpful to the other clients, which explains why the Oracle model performs the best.

---

> ### Author Response · Authors · 2022-08-07
> **The end of the discussion phase is approaching.**
>
> Dear Reviewer we3U,
>
> We sincerely appreciate your positive comments that the personalized weight aggregation with functional embeddings for identifying similar subgraphs is novel, the proposed method can handle the missing edge and community detection challenges, and our FED-PUB achieves superior performances against relevant baselines in more communication-efficient and privacy-preserving ways.
>
> However, you pointed out two weaknesses: the incompatible knowledge issue between subgraphs from different communities is unclear, and the random partitioning setting should be tested to further validate our method. To this end, in the responses below, we have clearly shown that the incompatible knowledge issue is severe in subgraph FL tasks with additional experimental results. Also, we have provided the results on the suggested random graph partitioning setting, where our FED-PUB significantly outperforms other baselines with a large margin.
>
> In the end, we are confident that we have fully addressed all of your comments/concerns in the initial responses. Therefore, since the discussion phase will close soon, could you please go over our complete responses? Please let us know if you have anything else that we should address.
>
> Best regards, Authors

---

> ### Author Response · Authors · 2022-08-09
> **A gentle reminder for Reviewer we3U**
>
> Dear Reviewer we3U,
>
> We believe that we have satisfactorily addressed all your two concerns: we clarify incompatible knowledge collapse issues and provide additional experimental results on the random graph partitioning you requested, in the initial response comments below. While the interactive discussion period between authors and reviewers has passed, we would like to gently remind you about our responses as well as the revision of our paper.
>
> We believe that incorporating your constructive comments and our discussion on them into our paper has significantly strengthened our work, and we would like to politely request you to go over our responses and then reflects our clarifications and revisions into your rating if they are satisfactory. We thank you again for your time and efforts in reviewing our paper.
>
> Best regards, Authors

---

### Official Review · Reviewer_rAFu · 2022-07-11

**Rating:** 6
**Confidence:** 4
**Soundness:** 3 good
**Presentation:** 3 good
**Contribution:** 3 good

**Summary:**

The paper deals with Federated Learning (FL) for graphs. Specifically, the authors investigate FL for heterogeneous graph distributions that can arise when the local graphs belong to different communities within the larger global graph. To tackle the threat of knowledge collapse of the server's GNN model, the introduced method, FED-PUB, tries to identify interrelated local graphs through functional embeddings and aims for their joint improvement. A weighted averaging on the server-side and locally learned weight masks achieve the combined local improvement. The authors evaluate FED-PUB on six different datasets.

**Questions:**

- The results for FedSage are worse compared to the original publication. How does FED-PUB perform for the subgraph division from [43](<https://proceedings.neurips.cc/paper/2021/hash/34adeb8e3242824038aa65460a47c29e-Abstract.html>)?
- The functional embeddings on random graphs, what are the failure cases of this approach? What if the graph distribution of these random graphs is off? SBM surely does not capture real-world graph properties. How does this translate?
- What is the effect of local graph size vs. heterogeneity and how well are the chosen graphs suited for/ support the presented method.
- Why is it not enough to average parameters according to similarity (Eq.4)? Please clarify the importance of local masks $\mu$ (4.2)?

Minor:

- Can you quantify to what degree training runs faster using local masks $\mu_k$?
- What is the influence of the scaling parameter $\tau$ (Eq.4)?
- Why this specific choice of $S(i,j)=\frac{h_i h_j}{\lVert h_i \rVert \lVert h_j \rVert}$ (Eq.3)?
- How are communication costs computed, cf. 52?
- What does the colouring indicate for Fig.6a "Missing edges"?

**Limitations:**

-

**Strengths And Weaknesses:**

Strengths:

- Combination of FL with the graph-theoretical concept of communities
- An improved design to ensure data privacy and security via functional embeddings of random graphs
- Providing a different perspective on the graph FL problem by aiming to improve local graph models

Weaknesses/Open Aspects:

- Please provide graph statistics to investigate the diversity of your evaluation settings across the six datasets
- Please elaborate on the theory of network homophily and the specific statements you build your arguments on, cf. citation [31]
- Please clarify the necessity of personalized sparse masks and quantify the drift to the local training distribution. Please provide insights for the specific choice of $\lambda_2$.
- For a more transparent comparison, the authors should analyse FED-Pub on the graph partitioning presented in [43]

Minor:

- There is a gap between the provided real-world motivation and the artificially constructed client graphs. Please give some insight into how well the shown setting translates to real scenarios.

### Originality

The topic of FL for graphs has been picked up only recently (most papers are from 2020 and 2021). Within this evolving research field, the authors introduce the challenge of appropriately dealing with heterogeneous graphs on the client-side. The authors combine graph FL with the well-defined notion of community detection from graph theory to address heterogeneous local graphs.

### Quality

- The authors motivate their work well, and the shown evaluation fits their claims
- The methods used are appropriate and clear.
- The presented work is complete with multiple references to the supplement materials

Negatives:

- The claims on community detection and network homophily should be supported with additional theory. The authors, however, provide good intuition why their introduced setting and evaluation are valid.
- The authors lack the chance to discuss weaknesses in the paper's main text.

### Clarity

- The paper is well written
- The authors present their work well structured
- The test is mostly self-contained

Minor:

- Introduction to GNNs is relatively shallow
- The authors exaggerate at times, "Seemingly impossible task" (4.1), "an even more serious issue" (1.), and "crucial drawback"
- Cite the inspiration paper [17] earlier

### Significance

- Dealing with FL, this work is essential for areas where data is not publically available due to privacy constraints.
- The benchmark for overlapping and non-overlapping subgraphs is new and can inspire future evaluations for graph FL
- The connection to graph theory (community) is valuable

Negatives:

- For a more transparent comparison, the authors should analyse FED-Pub on the graph partitioning preneted in [43]

---

> ### Author Response · Authors · 2022-07-31
> **Initial Response (5/5) to Reviewer rAFu**
>
> **Question 14:** How well are the chosen graphs suited for the presented method?
>
> **Answer:** Our chosen graphs are suited for the presented task and methods in two different aspects. Before explaining the specific reasons, please note that, in this work, we aim to deal with challenges in heterogeneity, where subgraphs from different communities have opposite properties, with our personalized weight aggregation and personalized parameter masking schemes. Then, since real-world graphs that we used for experimental comparisons all have the community structures: citation and product networks both have the communities of categories, they are suited for verifying the presented task and methods. Also, to construct the experimental settings of personalized subgraph FL, we partition the global graph with graph partitioning algorithms based on community detection. Therefore, the partitioned graphs have also community structures, and such community properties are suited for demonstrating the effectiveness of our methods.
>
> ---
>
> **Question 15:** Can you quantify to what degree training runs faster using local masks μ_k?
>
> **Answer:** Theoretically, we can train as much faster as the number of masked parameters. In other words, if we can mask 50% of the parameters, then the training will be two times faster than the full parameter training in terms of FLOPs. However, as described in Lines 247-248, to implement this sparsity in neural network training, the hardware that supports the zero-skipping operations is required, and the actual training speeds depend on such hardware performance. Note that analyzing such hardware performance goes beyond the scope of our work, therefore, which we leave as future work.
>
> ---
>
> **Question 16:** What is the influence of the scaling parameter τ (Eq.4)?
>
> **Answer:** τ value in Equation 4 influences how much each model receives the weights from the weights of the other similar/dissimilar subgraphs. For extreme cases, when we set the τ value as 0, the weight aggregation scheme is the same as the FedAvg model, since we aggregate all model parameters with equal weights. On the other hand, when we set the τ value as $\infty$, each model does not receive model weights from the other clients, and only trains with its own parameters (i.e., the Local model). Thus, if we would like to share weights largely among similar subgraphs, then the τ value should be set lower. On the other hand, if we would like to localize each model without parameter sharing, the τ value should be set higher.
>
> ---
>
> **Question 17:** Why this specific choice of $S(i,j)=\frac{h_i \cdot h_j}{‖hi‖‖hj‖}$ (Eq.3)?
>
> **Answer:** This is because cosine similarity is widely used to calculate the similarity between vectors in the embedding space.
>
> ---
>
> **Question 18:** How are communication costs in Figure 8 computed?
>
> **Answer:** As described in Lines 324-327, we measure communication costs by the sum of the number of active parameters (neurons) transmitted from client-to-server and server-to-client. Note that, regarding the client-to-server cost of our FED-PUB, we consider both the numbers of active neurons in the sparsified weight $\theta_k$ and the sparse mask $\mu_k$. On the other hand, regarding the server-to-client cost, we only consider the transmission cost of the sparsified weight $\bar{\theta}_{k}$ obtained from the personalized weight aggregation. Thus, since our FED-PUB transmits only the sparsified weights instead of the full weights, ours has more efficiencies in communication against FedAvg.
>
> ---
>
> **Question 19:** What does the coloring indicate for Fig.6a "Missing edges"?
>
> **Answer:** If there exist lots of missing edges between two subgraphs, the blue color for those two subgraphs becomes darker. Specifically, by comparing the number of edges in two subgraphs before/after graph partitioning with the METIS algorithm described in Lines 265-270, we can measure the number of missing edges between them, which we represent as color in Figure 6 (i.e., the larger the missing edges, the darker the blue color).

---

> > ### Comment · Reviewer_rAFu · 2022-08-09
> > **Reply**
> >
> > Overall, my curiosity is satisfied with the authors' response. However, the authors did not update the paper yet. I am happy to increase my score in case the paper gets revised. Along the other changes that you plan for the revision, I would suggest to include the the reasoning for the METIS algorithm also in the main text of the manuscript. I consider this design choice very important and highlighting differences to [43] valuable for the research community.

---

> > > ### Author Response · Authors · 2022-08-09
> > > **Thank you for your response**
> > >
> > > We sincerely appreciate your thoughtful reviews and comments, and also your reply to our initial responses. We are happy to hear that your comments/curiosities are satisfied with our responses.
> > >
> > > As suggested, we have faithfully included all the constructive discussions that we had with reviewers in our main paper and the supplementary file. Moreover, following your concrete suggestion, we highlight the difference in graph partitionings between ours (i.e., METIS) and FedSage [43] (i.e., Louvain) in Section 5.1 of the main paper and Section C.2 of the supplementary file. Note that, due to the page limits, we had to move some of the details in the supplementary file. However, if one page is further allowed after acceptance of this paper, we are sure to describe such differences in more concrete ways in the main paper.
> > >
> > > We hope our revision further satisfies you. Also, we hope you take this into consideration for your rating, as you promise. Once again, thank you so much for your constructive comments, and thank you for engaging with us.

---

> ### Author Response · Authors · 2022-08-01
> **Initial Response (4/5) to Reviewer rAFu**
>
> **Question 12:** The randomly generated graph from SBM does not capture real-world graph properties, and more analyses on different graph distributions are required.
>
> **Answer:** As described in Line 212, since the server cannot access the local graph properties (e.g., local node features) due to privacy concerns, we randomly generate the graph from SBM where node features are initialized by the normal distribution. However, when calculating the functional embeddings, using the random node features has an advantage over using the existing node features from clients, since randomness does not yield any bias on the functional space, while particularly initialized node features from exiting nodes and their label distributions might have a bias toward particular subspaces. To experimentally validate whether random initialization is superior to using the existing node features, we compare various schemes used for calculating the functional embeddings: 1) SBM denotes the random graph generated from the SBM model like ours; 2) ER denotes the random graph generated from the Erdos-Renyi model; 3) One denotes the random graph having only one node; 4) Feature denotes the graph where nodes are initialized by the node features in the client. We then measure the performances of those four schemes by calculating the correlation coefficients between actual label distributions and generated similarities of subgraphs (i.e., the high value means that the similarities from the functional embeddings are similar to the label distributions) on the Cora dataset of non-overlapping and overlapping node scenarios with 20 clients, which are reported in the table below.
>
> | Graph | Overlapping | Non-Overlapping |
> | --- | --- | --- |
> | SBM | 0.937 | 0.810 |
> | ER | 0.920 | 0.712 |
> | One | 0.822 | 0.656 |
> | Feature | 0.897 | 0.632 |
>
> As shown in the table, compared to the One scheme that uses only one node for calculating the functional embeddings, SBM and ER schemes that use more large numbers of randomly initialized nodes can accurately capture the similarities between subgraphs. This result confirms that a sufficient amount of randomness is required to identify the model’s functional space. Also, compared to the Feature scheme that uses existing node representations for calculating the functional embeddings, SBM and ER random models show superiority in capturing similarities among subgraphs, which verifies that randomness might help obtain accurate functional embeddings of the models without bias.
>
> ---
>
> **Question 13:** What is the effect of local graph size vs. heterogeneity?
>
> **Answer:** If the local graph size increases, the heterogeneity issue becomes severe. To quantify the heterogeneity issue over varying local graph sizes, we first measure the distances of each subgraph to all the other subgraphs by calculating their Jenson-Shannon divergence of label distributions, and then report the median divergence value between subgraphs on the number of clients of 3, 5, 10, and 20. Note that if the number of clients is small, then the number of nodes and edges in their subgraphs are also small, as shown in Table 1 of the supplementary file.
>
> | # Clients | Cora | CiteSeer | PubMed |
> | --- | --- | --- | --- |
> | 3 | 0.520 | 0.437 | 0.373 |
> | 5 | 0.590 | 0.517 | 0.362 |
> | 10 | 0.606 | 0.541 | 0.392 |
> | 20 | 0.665 | 0.568 | 0.424 |
>
> We have conducted analyses on the Cora, CiteSeer, and PubMed datasets of the nonoverlapping node scenario, and then provided the results in the above table. As shown in the table, we can confirm that, when the number of clients is increased (i.e., the numbers of nodes and edges are decreased), the heterogeneity across clients becomes more severe and problematic for personalized subgraph FL, and thus is an important issue to tackle.

---

> ### Author Response · Authors · 2022-08-01
> **Initial Response (3/5) to Reviewer rAFu**
>
> **Question 5:** There is a gap between the provided real-world motivation and the artificially constructed client graphs. Please give some insight into how well the shown setting translates to real scenarios.
>
> **Answer:** Unfortunately, since real-world datasets for graph FL are not available due to privacy concerns, pioneer [43] and our works partition the given entire graph into different subgraphs with community detection algorithms. However, we believe that our experiment setting clearly reflects real-world scenarios. In particular, in realistic scenarios, similar clients (e.g., hospitals for the same sector) are likely to have similar nodes with many connections among them, while dissimilar clients (e.g., hospitals for different sectors) are likely to have nodes with opposite properties that are not connected to each other. Then, in this real-world example, subgraphs from similar clients are likely to belong to the community, and, in our experiment, we allow each client to have a subgraph belonging to the particular community by dividing the global graph based on its community structures, not based on any arbitrary random structures. Therefore, our motivation (i.e., a real graph has a community structure where each client is associated with the particular community) aligns with our experimental setup (i.e., each client has a subgraph that belongs to the particular community).
>
> ---
>
> **Question 6:** The authors lack the chance to discuss weaknesses in the paper's main text.
>
> **Answer:** Due to the space issue, we provide the limitations of our work in Section D of the supplementary file. If an additional one page is allowed after the acceptance, following your suggestion, we will discuss weaknesses in the main paper.
>
> ---
>
> **Question 7:** Introduction to GNNs is relatively shallow.
>
> **Answer:** Thank you for pointing this out. Due to the space limits, we will add more explanations about GNNs in the next revision if additional space is allowed.
>
> ---
>
> **Question 8:** The authors exaggerate at times, "Seemingly impossible task" (4.1), "an even more serious issue" (1.), and "crucial drawback".
>
> **Answer:** Thank you for pointing this out. Our original intention was that we would like to emphasize the importance of challenges prevalent in subgraph FL tasks, however, since they might feel exaggerated, we will tone down on those parts accordingly in the revision.
>
> ---
>
> **Question 9:** Cite the inspiration paper [17] earlier.
>
> **Answer:** Thank you for your suggestion. We will cite the inspiration paper [17] earlier (e.g., Introduction section) in the next revision.
>
> ---
>
> **Question 10:** The results for FedSage [43] are worse compared to the original publication.
>
> **Answer:** The reported results in FedSage [43] are not comparable, as the datasets and evaluation setups are different between FedSage and ours. At first, as discussed in answers to Question 4, FedSage uses the Louvain algorithm while we use the METIS algorithm to partition the global graph into different subgraphs, thus the training and test datasets are completely different. Furthermore, regarding the evaluation setup, FedSage [43] measures the performance on the entire graph (i.e., not the subgraphs) with the global model averaged by all client parameters. However, we measure the average performance on the individual subgraphs with their own personalized model parameters, as we deal with personalized scenarios.
>
> ---
>
> **Question 11:** Regarding the functional embeddings on random graphs, what are the failure cases of this approach?
>
> **Answer:** In experiments, we have not observed clear failure cases of our graph functional embeddings. However, one worthwhile point to share is that the scaling hyperparameter τ should be set differently across overlapping and nonoverlapping node scenarios as described in Section B.3 of the supplementary file, since the scales of similarities of those two settings are different. Thus, we suspect that our tuned hyperparameters τ may not be optimal if we deal with totally different evaluation setups: neither non-overlapping nor overlapping.

---

> ### Author Response · Authors · 2022-08-01
> **Initial Response (2/5) to Reviewer rAFu**
>
> **Question 3.1:** Please clarify the necessity of personalized sparse masks and quantify the drift to the local training distribution.
>
> **Answer:** We already clearly showed that personalized sparse masks are significantly helpful when all the **subgraphs are disjoint** (i.e., there exist extreme distribution shifts between subgraphs), compared to the overlapping node scenario in Figure 7, showing the necessity of the weight masking scheme for subgraph FL tasks. However, to further quantify how the weight masking scheme benefits the node classification task, we have additionally measured how much distributional shifts exist among the subgraphs, as suggested.
>
> Specifically, to quantify the necessity of personalized sparse masks, we have measured the label differences (i.e., distributional shifts) between subgraphs with the Jenson-Shannon divergence, in which the minimum and maximum values are 0 and 2, respectively, on the Cora dataset with 20 different clients over the overlapping and the non-overlapping scenarios. Then, we observe that the distance (i.e., divergence value) among the subgraphs within the same community is 0.384 while the distance among the subgraphs belonging to different communities is 0.639 for the non-overlapping node scenario. On the other hand, the distance among the subgraphs within the same community is 0.047 while the distance among the subgraphs belonging to different communities is 0.528 for the overlapping node scenario. Thus, from the fact that heterogeneity of subgraphs within the same community is extremely larger in the non-overlapping setting (0.384) compared to the overlapping setting (0.047), personalized weight aggregation might not be enough in disjoint subgraph FL problems, and, as shown in Figure 7, weight masking scheme is clearly required to selectively utilize and train only on the relevant parameters to each local task.
>
> Furthermore, besides the above advantage of weight masks on disjoint subgraph scenarios, parameter sparsifications with sparse masks can further yield computation and communication efficiencies as described in Lines 245-249 with experimental results in Figure 8. Note that efficiencies are key factors in FL (e.g., considering the case where we often do federate learning on edge devices), which is, however, getting less attention in graph FL yet we have explored them.
>
> ---
>
> **Question 3.2:** Please provide insights for the specific choice of λ2.
>
> **Answer:** As described in Section B.3 of the supplementary file, we simply set λ2 as 0.001 which is the same as the λ1 value for all experiments. This is because we empirically observed this value to work well (i.e., preventing local divergence as shown in Figure 9) across different settings. However, we also varied λ2 in Figure 1 of the supplementary file, and the results show that the performance is not sensitive across different λ2 values.
>
> ---
>
> **Question 4:** The authors should analyze FED-PUB on another graph partitioning setting presented in [43].
>
> **Answer:** Thank you for your suggestion. However, there is a drawback when using the Louvain algorithm presented in [43], rather than using the METIS algorithm as ours, for subgraph FL scenarios. Specifically, since the Louvain algorithm cannot specify the number of graph partitions, the number of subgraphs on the CiteSeer dataset is 38, where three of them have less than ten nodes. Then, based on those 38 disjoint subgraphs, to generate the particular number of clients (e.g., 10), [43] randomly merges the different subgraphs without considering their graph properties. Therefore, even though each partitioned subgraph has its unique structural role/characteristic, the reconstructed 10 subgraphs from the original 38 subgraphs have mixed properties (i.e., two incompatible subgraphs could be merged), which is suboptimal. However, as described in Lines 265-266, the METIS algorithm that we used can specify the number of partitions, thus more appropriate for making the experimental settings for subgraph FL.
>
> However, as suggested, we have additionally conducted experiments with the Louvain graph partitioning algorithm [43], on Cora, CiteSeer, and PubMed datasets with the number of clients as 10, and then reported the results in the table below. Then, the results show that our FED-PUB consistently outperforms all the other baselines on another graph partitioning setting, thus the effectiveness of our FED-PUB becomes more obvious.
>
> | Model | Cora | CiteSeer | PubMed |
> | --- | --- | --- | --- |
> | Local | 78.56 ± 0.27 | 64.06 ± 0.09 | 84.07 ± 0.17 |
> | FedAvg | 71.83 ± 0.40 | 69.23 ± 0.71 | 82.47 ± 0.32 |
> | FedProx | 72.09 ± 0.29 | 67.66 ± 0.97 | 82.68 ± 0.34 |
> | FedPer | 80.13 ± 0.50 | 66.28 ± 1.22 | 85.02 ± 0.23 |
> | FedGNN | 76.59 ± 0.66 | 61.21 ± 1.46 | 82.67 ± 0.26 |
> | FedSage | 72.20 ± 0.60 | 68.40 ± 0.61 | 82.76 ± 0.09 |
> | GCFL | 78.55 ± 0.38 | 64.20 ± 0.31 | 84.62 ± 0.31 |
> | FED-PUB (Ours) | **82.68** ± 0.13 | **69.45** ± 0.75 | **86.20** ± 0.11 |

---

> ### Author Response · Authors · 2022-08-02
> **Initial Response (1/5) to Reviewer rAFu**
>
> We sincerely thank you for your constructive and helpful comments. We appreciate your positive comments, which praise the strengths of our work in terms of significance, originality, quality, and clarity. We initially address all your concerns below:
>
> ---
>
> **Question 1:** Please provide graph statistics to investigate the diversity of your evaluation settings across the six datasets.
>
> **Answer:** We already provided the graph statistics, such as the number of nodes, edges, and classes, in Table 1 of the supplementary file. However, we have further analyzed the additional graph statistics, namely the clustering coefficients and the heterogeneities for subgraphs, and here we provide them for the Cora dataset of the non-overlapping node scenario in the table below.
>
> | Dataset | 5 clients | 10 clients | 20 clients |
> | --- | --- | --- | --- |
> | # Classes | 7 | 7 | 7 |
> | # Nodes | 497 | 249 | 124 |
> | # Edges | 1,866 | 891 | 422 |
> | Clustering coefficient | 0.250 | 0.259 | 0.263 |
> | Heterogeneity | 0.590 | 0.606 | 0.665 |
>
> Note that clustering coefficients, which are the measure of how much nodes tend to cluster together, are calculated by the average of clustering coefficients [A] of all nodes. On the other hand, we calculate the heterogeneities by measuring the median Jenson-Shannon divergence of label distributions between all pairs of subgraphs. We will add those statistics of the real-world datasets in Table 1 of the supplementary file, in the revision.
>
> [A] Watts and Strogatz. Collective dynamics of 'small-world' networks. Nature 1998.
>
> ---
>
> **Question 2.1:** Please elaborate on your statements about the theory of network homophily and community detection.
>
> **Answer:** While we explain those two concepts and their connection in Lines 161-164, here we further clarify our statements. At first, from the definition of network homophily [31], nodes with similar properties are more likely to connect to each other than dissimilar ones. Then, based on this definition, similar nodes are densely connected, which then form a community from the definition of the community [33, 9, 32] (i.e., the number of edges within the same community is larger than the number of edges estimated by the random model). In this regard, for the subgraph FL problem that we target, we specify two facts from the graph theories above that it is important to capture the community structures to promote joint improvements among subgraphs having similar properties, and to handle the incompatible knowledge issues between subgraphs in different communities. Therefore, to summarize, theoretical concepts of network homophily and network community motivate us to find the challenges on subgraph FL, and, from this, we define our novel personalized subgraph FL problem represented in Equation (2).
>
> ---
>
> **Question 2.2:** The claims on community detection and network homophily should be supported with additional theory, while the authors provide good intuition why their introduced setting and evaluation are valid.
>
> **Answer:** Thank you for your comment, however, we strongly believe that, in our work, we not only clearly introduce our ideas based on graph theories as explained above (the answer for Question 2.1), but also sufficiently validate our ideas with various experimental results. Also note that, for deep networks, it is challenging to analyze the functional behavior of every deep layer, and then invent a new theoretical justification. For this reason, many recent works on deep FL (e.g., [2, 40]) do not provide theoretical analyses, while they are well received due to their task-level ideas and sufficient empirical verifications. Thus, to summarize, we believe that we already sufficiently motivated our subgraph FL tasks based on theoretical concepts of network homophily and community and that further analyzing the theoretical behavior of our deep FL methods goes beyond the scope of our work, thus we leave it as future work.

---

> ### Author Response · Authors · 2022-08-07
> **The end of the discussion phase is approaching.**
>
> Dear Reviewer rAFu,
>
> We sincerely appreciate your positive comments that our work is well motivated by graph-theoretical concepts of communities, the proposed methods are appropriate to handle suggested subgraph FL problems in privacy-preserving ways, the benchmark experiments on overlapping and non-overlapping subgraphs are valuable, and the experimental evaluations support our claims well, with high clarity in writing.
>
> In the responses below, we have made every effort to address all your concerns/comments. To mention a few, we have provided the additional experimental results under the suggested experimental settings of [43] and the additional dataset statistics you requested. Also, we have clarified our statement on network homophily and network communities with their theoretical connections, and experimentally validated the necessity of our sparse weight masks.
>
> As the discussion phase will close soon, could you please go over our complete responses? While we strongly believe that we have sufficiently addressed all your comments in the initial responses, please let us know if you have anything else that we should address.
>
> Best regards, Authors

---

### Meta-Review · Area_Chair_N5Qi · 2022-08-24

**Recommendation:** Reject
**Confidence:** Certain

**Metareview:**

The author(s) present(s) a new subgraph federated learning approach to learn a single GNN model that computes embeddings based on the relationship between local graphs. This approach goes beyond the previous approaches that consider the local subgraphs separately.

 The paper is interesting and present some novel ideas but it should address two fundamental critiques by the reviewers before acceptance. In particular the authors should:
- clarify the novelty of their approach(especially considering the missing discussion with previous work)
- improve the experimental section. The current results are interesting but not fully convincing and the experimental section could be improved significantly by including additional settings suggested by the reviewers

Overall, the paper is interesting but it is not ready for publication at this point

**Award:**

No

---

### Decision · Program_Chairs · 2022-09-14

Reject